# MTCH2 promotes BAX and BAK self-assembly and apoptotic pore growth

Hector Flores-Romero [1,2,3,9], Aida Pena-Blanco[4,9], Jonas Aufdermauer [1,5,9], Shashank Dadsena[1,6], Philip Neubert [7], Lisa Hohorst [1,5], Eileen Cors[8], Cristiana Zollo[1], Alvaro Larrañaga-SanMiguel[2], Jone Zaldunbide[2], Maria Nenchova [1], Yasmin Carvalho-Schaefer [1], Thomas Langer[8], Georg Häcker[7] & Ana J. Garcia-Saez [1,5] ✉

During apoptosis, the BCL-2 family members BAX and BAK oligomerize and form a pore to mediate the decisive step of mitochondrial outer membrane permeabilization. However, the contribution of additional cellular components to apoptotic pore dynamics remains poorly understood. Here we map the protein environment of the apoptotic pore using in situ proximity labeling and identify the mitochondrial carrier homolog protein MTCH2 localizing nearby BAX and BAK assemblies specifically under apoptotic conditions. We show that cells lacking MTCH2 exhibit delayed BAX and BAK oligomerization at the single-particle level, which can be rescued by addition of lysophosphatidic acid. Accordingly, MTCH2 depletion decreases not only apoptosis sensitivity but also sublethal mitochondrial permeabilization during bacterial infection, mitochondrial DNA release into the cytosol and cGAS–STING activation under impaired caspases. Our findings uncover a key role of MTCH2 in promoting BAX and BAK high-order assembly with functional consequences for apoptotic pore growth and downstream responses.

Apoptosis is a form of regulated cell death required for key biological processes, such as organ shaping during embryo development, immune system function or tissue homeostasis, by eliminating damaged or superfluous cells[1]. Dysregulation of apoptosis is frequently associated with human diseases, such as cancer and neurodegeneration[2–4]. In the intrinsic apoptosis pathway, widespread mitochondrial outer membrane (MOM) permeabilization (MOMP) is considered a decisive step in the cell's commitment to death. It enables the release of apoptotic factors (such as cytochrome c (cyt c) or SMAC) into the cytosol, triggering apoptotic caspases activation and cell death[5]. The apoptotic pore also exposes other mitochondrial contents to the cytosol, including mitochondrial DNA (mtDNA), which can induce activation

of the cGAS–STING pathway under low caspase activity, modulating the inflammatory outcome of apoptosis[6–9].

The proapoptotic proteins of the BCL-2 family, BAX and BAK, are key mediators of MOMP[10,11]. Each of them alone exhibits membrane-pore-forming activity in vitro, supporting their individual role as a minimal machinery for MOMP[12]. Whereas BAX is cytosolic and BAK is homogeneously distributed at the MOM under healthy conditions, during apoptosis, they activate and accumulate as discrete foci at mitochondria, which here we call apoptotic foci[1,13,14]. There, BAX and BAK coassemble into high-order oligomers that organize into a heterogeneous distribution of structures including lines, arcs and rings[13–18]. They directly mediate the opening of the apoptotic pore, which is also

[1]Institute for Genetics and CECAD, University of Cologne, Cologne, Germany. [2]Achucarro Basque Center for Neuroscience, Leioa, Spain. [3]Ikerbasque, Basque Foundation for Science, Bilbao, Spain. [4]Interfaculty Institute of Biochemistry, University of Tübingen, Tübingen, Germany. [5]Max Planck Institute of Biophysics, Frankfurt, Germany. [6]Manipal School of Life Science, Manipal Academy of Higher Education, Manipal, India. [7]Institute of Medical Microbiology and Hygiene, Medical Center, University of Freiburg, Faculty of Medicine, Freiburg, Germany. [8]Max Planck Institute for Biology of Ageing, Cologne, Germany. [9]These authors contributed equally: Hector Flores-Romero, Aida Pena-Blanco, Jonas Aufdermauer. ✉e-mail: ana.garcia@biophys.mpg.de

rimmed by lipid molecules and grows over time[6,13,14]. In the cellular context, the self-assembly of BAX or BAK is modulated by the other[14] and likely by yet unknown factors. Unraveling the molecular constituents and regulators of the apoptotic pore is, thus, key to understanding how MOMP is dynamically regulated, which is of relevance for identifying novel clinical targets[19].

The study of BAX and BAK interaction partners at the apoptotic pore has proven to be a major challenge given the complexity and dynamic nature of the structures they form. Previous studies usually involved cell lysis with detergents[20,21], which destroy the native membrane environment of BAX and BAK macromolecular complexes, resulting in their disassembly and potentially in the loss of interactions. Consequently, little is known about the cellular contributors to the apoptotic pore beyond BAX and BAK.

To overcome these limitations, here we used APEX2 proximity-dependent labeling coupled with mass spectrometry (MS) analysis for in situ detection of proteins neighboring BAX and BAK in their native environment. We identify several mitochondrial proteins enriched in the proximity of BAX and BAK specifically during apoptosis, with MTCH2 chief among them. MTCH2 is a poorly understood mitochondrial carrier homolog protein that has been linked with apoptosis, mitochondrial fusion and lipid metabolism and can act as a protein insertase[22–32]. Quantitative analysis at the single-particle level reveals that MTCH2 promotes BAX and BAK supramolecular oligomerization, whose reduced assembly in absence of MTCH2 can be counterbalanced by treatment with lysophosphatidic acid (LPA). We find that this function of MTCH2 on the apoptotic pore impacts mtDNA release and cGAS–STING pathway activation. Consistently, we also show that MTCH2 has a role in sublethal apoptosis and subsequent DNA damage in *Helicobacter pylori* infection. Our findings reveal that apoptotic pore growth can be modulated by other proteins and identify MTCH2 as a key regulator, with functional consequences for inflammatory and sublethal cell death signaling.

## Results

### In situ proximity labeling for identification of components of the apoptotic foci

In situ proximity labeling coupled with MS allows exploring spatially restricted proteomes, as well as the composition of protein complexes. The engineered ascorbate peroxidase APEX2, with fast labeling kinetics[33], is suitable for studying fast cellular processes, such as apoptosis, typically executed within 1 h. In the presence of biotin-phenol (BP) and hydrogen peroxide ($H_2O_2$), APEX2 generates biotin-phenoxyl radicals, covalently tagging proximal (-10 nm) endogenous proteins in their native cellular environment[34]. Subsequent enrichment of biotinylated proteins using streptavidin beads allows for their identification by MS.

To identify components of the mitochondrial apoptotic pore, we generated APEX2-tagged fusion constructs with BAX and BAK. We also included the mitochondrial fission regulator DRP1, which presents mixed cytosolic and mitochondrial localization in healthy cells but colocalizes with BAX and BAK during apoptosis[35]. In the constructs, the APEX2 region is preceded by a FLAG tag for immunostaining and western blot (WB) (Fig. 1a). We reasoned that potential apoptotic pore constituents would be biotinylated specifically during apoptosis by all three constructs, whereas APEX2-BAX2, APEX2-BAK and APEX2–DRP1 samples from healthy cells would provide negative controls for unspecific cytosolic proteins, mitochondrial proteins and both, respectively (Fig. 1b).

We next confirmed the biotinylation pattern and the subcellular mitochondrial localization of the constructs during apoptosis (Fig. 1c,d). We performed biotinylation reactions in wild-type (wt) HeLa cells transiently expressing APEX2–BAX, APEX2–BAK and APEX2–DRP1 treated or not with staurosporine (STS) for apoptosis induction. As expected, in labeling controls without BP or APEX2 expression, only endogenously biotinylated proteins were detected. After STS treatment, all three APEX constructs localized to mitochondria and distributed into discrete foci (Fig. 1d), confirming that the fusion tag did not notably alter subcellular localization. Biotinylation was confined in the vicinity of the MOM-located BAX, BAK and DRP1 (Fig. 1d). The MOM of healthy cells contains porins allowing the diffusion of the BP radical, which may result in the biotinylation of intermembrane space (IMS) and mitochondrial inner membrane (MIM)-resident proteins in healthy cells[34]. These mitochondrial compartments are fully accessible upon MOMP.

After transfection with the corresponding APEX2-containing construct, treatment or not with STS and live-cell biotinylation, followed by streptavidin-based enrichment of biotinylated proteins, samples were digested with trypsin, dimethyl-labeled and analyzed by liquid chromatography (LC)–MS/MS in three independent quantitative MS experiments (Fig. 1e). Each experiment involved three conditions: untreated (CTRL) and apoptotic (STS) for the same APEX2 construct and STS APEX2–BAX as a common reference in all samples. For the APEX2–BAX construct, we included STS APEX2–BAK (Fig. 1e). For each construct, we analyzed biotinylated proteins enriched in apoptosis by comparing the $\log_2$(STS/CTRL) values. We set a cutoff ratio at the 90th percentile of the normal distribution of $\log_2$(STS/CTRL), above which we selected the top candidates ($\log_2 > t_{0.9}$; first criterion). Independent of the APEX2 fusion construct used in the untreated sample, we observed a strong correlation in the STS/CTRL values across experiments (Supplementary Fig. 1a–c), indicating similar enrichment patterns. APEX2-fused proteins were excluded because of overexpression.

We reasoned that proteins located at apoptotic foci would be similarly enriched for BAX, BAK and DRP1 in apoptosis and, therefore,

**Fig. 1 | In situ proximity labeling of the apoptotic pore protein environment. a**, Representation of APEX2 constructs. FLAG–APEX2 was fused to the N-terminal region of BAX, BAK and DRP1. **b**, Scheme of the expected localization of the different APEX2 fusion proteins (green) and the extension of APEX2-catalyzed biotinylation (yellow) upon addition of BP and $H_2O_2$ in healthy and apoptotic cells. **c**, Streptavidin blot of the APEX2 constructs for BAX, BAK and DRP1 after APEX2-catalyzed biotinylation. Top and bottom blots correspond to streptavidin–horseradish peroxidase (Strep–HRP) and Ponceau staining, respectively. Apoptosis was induced with STS for 3 h before the biotinylation reaction. Representative immunoblot of three independent experiments. **d**, Confocal fluorescence imaging of APEX2–BAK, APEX2–BAX and APEX2–DRP1 constructs in HeLa cells treated with STS for 3 h. Cells were stained with anti-FLAG–Alexa Fluor 488 (green) to detect the fusion constructs and anti-Strep–Alexa Fluor 555 (magenta) to detect biotinylated proteins. Mitochondria were labeled with mitoBFP (blue). Bottom images show zoomed selected areas from the top images. Images are representative of three independent experiments. Scale bars, 10 μm for top images and 5 μm for zoomed-in images. **e**, Design of

proteomics experiment. Three experiments with three conditions each were performed. Each experiment includes one untreated (CTRL) and two apoptotic (STS) samples with different combinations of the APEX2 fusion constructs. APEX2–BAX under apoptotic conditions was used in all experiments as a reference sample. For each experiment, $\log_2$(STS/CTRL) values (left histogram) allowed the identification of proteins changed in apoptosis, whereas $\log_2$(STS/STS) values (right histogram) provided information about proteins equally enriched between the indicated samples in apoptotic conditions. We filtered for mitochondrial and apoptotic annotation to identify top candidates for constituents and/or regulators of the apoptotic pore. **f**, Scatter plot showing $\log_2$(BAX STS/BAX CTRL) versus $\log_2$(BAK STS/BAX STS). Proteins inside the yellow rectangle were enriched in apoptosis (first criterion), while proteins inside the blue rectangle corresponded to equally enriched proteins in apoptosis conditions (second criterion). The intersected area was magnified and mitochondrial proteins involved in apoptosis according to the GOBP were labeled. **g**, Same analysis as in **f**, but for the APEX2–BAK experiment. **h**, Same analysis as in **f**, but for the APEX2–DRP1 experiment.

included candidates with $\log_2(STS/STS)$ values within the arithmetical mean ± s.d. of all $\log_2(STS/STS)$ values (second criterion) (Fig. 1e). Proteins fulfilling both criteria were filtered for mitochondrial annotation (third criterion) and apoptosis involvement (fourth criterion) using the Gene Ontology (GO) cellular component (GOCC) database[36]. Proteins satisfying all four criteria were considered as candidates for constituents or regulators of apoptotic foci (Fig. 1f–h).

The top candidates from experiment 1 (CTRL:BAX, STS:BAX and STS:BAK) are shown in Fig. 1f. We identified 26 proteins satisfying the first and second criteria, among which 15 were annotated as mitochondrial in GOCC and 10 were involved in apoptotic processes according to GO biological process (GOBP). The same analysis of experiment 2 (CTRL:BAK, STS:BAX and STS:BAK) identified 16 proteins, including eight with mitochondrial and five with both mitochondrial and

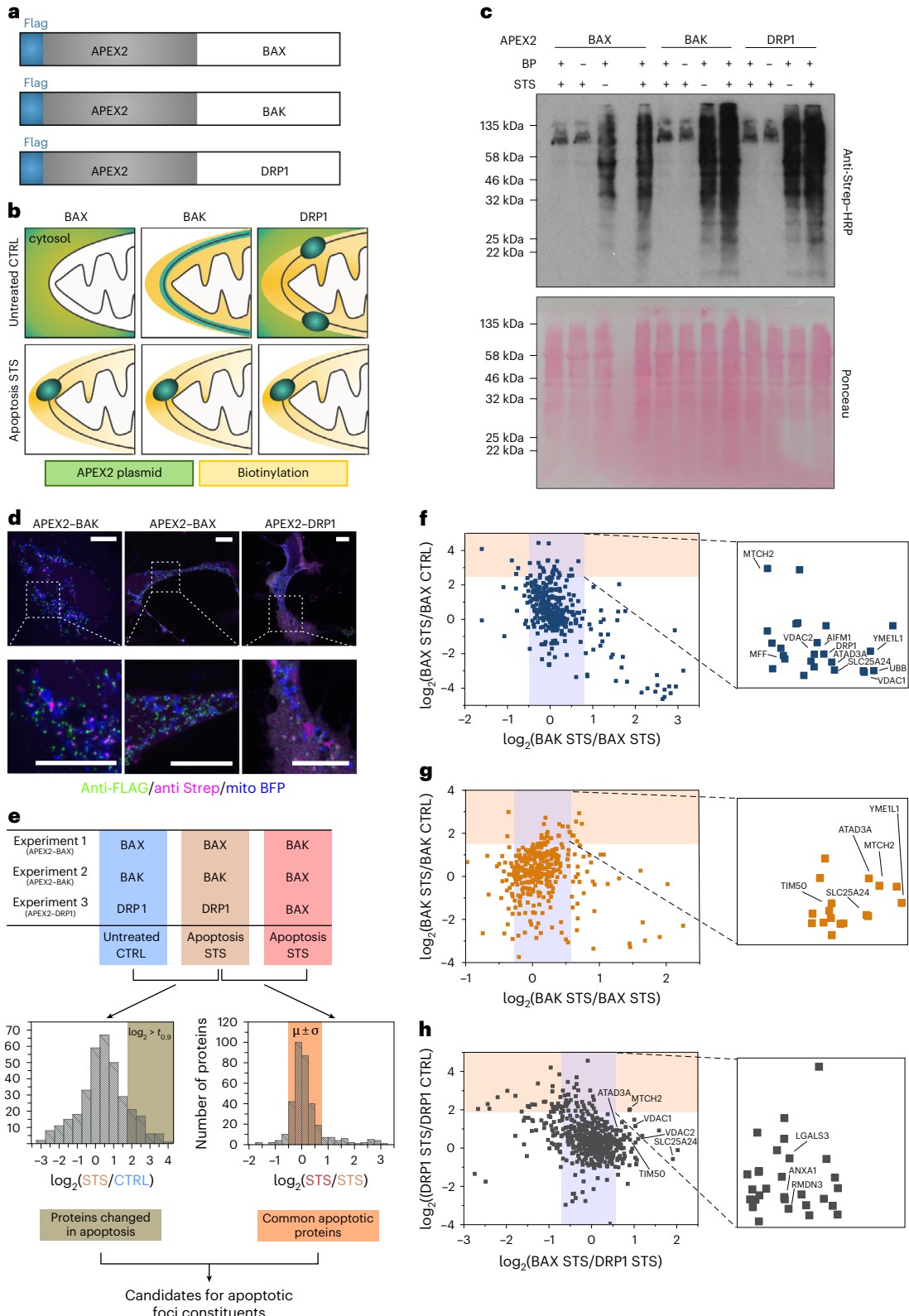

apoptotic annotation (Fig. 1g). Four of these five proteins were also identified in experiment 1 (MTCH2, YME1L1, ATAD3A/B and SLC25A24), indicating the similar protein environment of BAX and BAK during apoptosis. DRP1 was also enriched but did not fulfill the first criterion. In experiment 3 (CTRL:DRP1, STS:DRP1 and STS:BAX), 31 proteins satisfied the first and second criteria but only five were mitochondrial, among which three were related to apoptosis: LGALS3, ANXA1 and RMDN3 (Fig. 1h). BAK was not detected by MS. BAX and BAK hits were selected in APEX2–DRP1, displaying high $\log_2(BAX_{STS}/DRP1_{STS})$ values (labeled proteins in the magnified area of Fig. 1h), suggesting closer proximity to BAX than to DRP1. Among them, MTCH2 was also significantly enriched.

Figure 2a shows a Venn diagram of proteins with $\log_2(STS/CTRL)$ showing a >1.4-fold increase during apoptosis in all samples. The BAK sample presented fewer apoptosis-enriched candidates than BAX and DRP1, perhaps because BAK is constitutively mitochondrial. About two thirds of the BAK candidates were also found in the BAX samples, whereas DRP1 samples presented more different enriched proteins. These results suggest that BAX and BAK occupy a more similar subcellular environment than DRP1 during apoptosis, despite the colocalization of all three[14,35]. Consistently, Metascape analysis after applying the first and second criteria identified mitochondrion organization as the top nonredundant enrichment cluster for BAX and BAK but not DRP1 (Fig. 2b–d). On this basis and considering the key role of BAX and BAK in apoptosis, we focused on top candidates from the BAX and BAK samples.

Together, our results identify several proteins in close proximity to both BAX and BAK during apoptosis. MTCH2, ATAD3, YME1L and SLC25A24 emerged as top candidates. MTCH2 was also significantly enriched in apoptotic DRP1 samples and ranked first and second in BAX and BAK scatter plots of apoptosis versus untreated $\log_2$ protein ratios, respectively (Fig. 2e). Therefore, MTCH2 was selected for further investigation.

We validated the apoptosis-specific proximity labeling of MTCH2 detected by MS using WB. MTCH2 was detected in all three APEX2 constructs exclusively in apoptotic samples, supporting its close proximity to BAX, BAK and DRP1 during apoptosis (Fig. 2f). As an orthogonal approach, we used dimerization-dependent fluorescent proteins (ddFPs)[37,38]. In this method, a nonfluorescent self-quenched red chromophore (RA) and a nonfluorescent monomer (GB), which significantly enhances RA fluorescence upon their interaction, are fused to the two proteins of interest (Fig. 2g). We induced apoptosis with STS for 3 h in HeLa cells transiently expressing RA–BAX and MTCH2–GB fusion constructs, which appeared as fluorescent puncta at mitochondria of

apoptotic cells (Fig. 2h), supporting their spatial association during apoptosis. We further confirmed the proximity of endogenous MTCH2 and active BAX by proximity ligation assay (PLA) in U2OS cells treated with ABT-737 and S63845 (COMBO) (Fig. 2i,j). Together, these results place MTCH2 in close proximity to active BAX during apoptosis.

## MTCH2 deficiency delays BAX and BAK high-order oligomerization and uncouples it from MOMP

To investigate how MTCH2 affects BAX and BAK function at the apoptotic pore, we visualized the spatiotemporal dynamics of BAK subcellular distribution during apoptosis progression with confocal microscopy using U2OS cells lacking BAK (ΔBAK) and stably expressing GFP–BAK, deficient or not in MTCH2 (Fig. 3a–e and Supplementary Fig. 2a). In line with a previous study[29], mitochondrial depolarization, measured as TMRE signal loss (a proxy for MOMP), occurred approximately 30 min after apoptosis induction in both wt and MTCH2-knockout (KO) cell lines (Fig. 3b).

Strikingly, GFP–BAK reorganization from a homogeneous mitochondrial distribution in healthy cells into apoptotic foci upon COMBO treatment was visibly delayed in MTCH2-KO cells (Fig. 3a). This was quantified by calculating the temporal evolution of the s.d. of the GFP–BAK signal and its maximum intensity per pixel, which were significantly delayed in MTCH2-KO cells compared to wt cells (Fig. 3c,d). We also analyzed the lag time between mitochondrial depolarization and focus formation. Remarkably, while both events occurred concurrently in wt cells, TMRE loss clearly preceded BAK focus formation in MTCH2-KO cells (Fig. 3e).

We then analyzed BAK oligomerization over time during apoptosis at the single-particle level using photon-counting confocal microscopy combined with ratiometric analysis, as described previously[14]. Here, the number of BAX or BAK molecules in individual apoptotic foci can be quantified from the photon counts of the single particles using the GFP-tagged nuclear pore complex component 96 (NUP96) as a standard. Consistently, MTCH2 depletion impaired BAK oligomerization, shown by both lower molecularity per particle and reduced number of apoptotic foci per cell (Fig. 3f).

As BAX and BAK coassemble during apoptosis into the same apoptotic foci[14], we transiently expressed GFP–BAX in wt and MTCH2-KO U2OS cells and quantified GFP–BAX assembly kinetics during apoptosis. Comparably to BAK, loss of MTCH2 delayed BAX reorganization into foci, giving rise to fewer foci with lower stoichiometry (Fig. 3g).

Together, these results show that the high-order assembly of BAX and BAK into apoptotic foci can be uncoupled from mitochondrial permeabilization during apoptosis progression and is compromised

**Fig. 2 | MTCH2 and BAX are in close proximity during apoptosis. a**, Venn diagram of proteins with $\log_2(STS/CTRL)$ showing a >1.4-fold increase during apoptosis from APEX2–BAX, APEX2–BAK and APEX2–DRP1. Common proteins in all three groups are listed. **b–d**, Metascape bar graph analysis showing top nonredundant enrichment clusters from biotinylated proteome of individual experimental groups fulfilling first and second criteria. Biological pathway enrichment cluster for APEX2–BAX proteome (**c**), APEX2–BAK proteome (**d**) and APEX2–DRP1 proteome (**e**). The horizontal axis represents the *P* values (<0.05) of GO terms on Metascape using an unpaired, one-sided enrichment test based on the cumulative hypergeometric distribution, equivalent to Fisher's exact test for overrepresentation. Mitochondria-related terms are labeled in blue, membrane-related terms are labeled in gray, infection-related terms are labeled in violet and other terms are labeled in green. **e**, Scatter plot of $\log_2$ of enriched protein ratios (A/B): dark blue, BAX STS/BAX control (proteins enriched nearby BAX in apoptosis); violet, BAK STS/BAX control (proteins enriched nearby BAK in apoptosis); light blue, BAK STS/BAX STS (proteins enriched similarly in BAX and BAX apoptotic samples). MTCH2 (yellow) is highly enriched in apoptosis for BAX and BAK and similarly enriched for both of them (ratio close to 1 or $\log_2$ 0). **f**, Immunoblots of streptavidin immunoprecipitation of APEX2–BAX (left), APEX2–BAK (middle) and APEX2–DRP1 from apoptotic HeLa cells expressing FLAG–APEX2–BAX, FLAG–APEX2–BAK and FLAG–APEX2–DRP1. Immunoblots

are representative of *n* = 2 independent experiments. Only regions of the gel with bands of interest are shown for clarity. Blots of BAX and BAK for FLAG–APEX2–BAX and FLAG–APEX2–BAK pulldowns were published previously[14]. Together, they represent the comparison of BAX, BAK, MTCH2 and DRP1 pulled down specifically under apoptotic conditions. **g**, Principle of ddFP analysis. RA only becomes fluorescent when it is in complex with GB. **h**, Representative confocal microscopy image of a BAX/BAK-DKO U2OS cell transfected with MTCH2–GB, RA–BAX and mitoBFP (magenta) after apoptosis induction. The interaction between BAX and MTCH2 (green dots) takes place in mitochondria upon apoptosis. Scale bar, 10 μm. Images are representative of two independent experiments. **i,j**, PLA for detecting proximity between MTCH2 and active BAX during apoptosis. **i**, Representative image of U2OS cells untreated or treated with 1 μM ABT-737 and S63845 (COMBO) and 10 μM QVDA. PLA signal reflecting active BAX:MTCH2 proximity is shown in green, mitochondria is shown in magenta and nuclei are shown in blue. The ROI corresponds to the white box. **j**, Quantification of the mean fluorescence intensity of the PLA signal from *n* = 3 independent experiments (>24 cells per condition). Small dots indicate individual cells, colored by experiment. Large dots represent the average per experiment. Statistical analysis was conducted using an unpaired, one-way analysis of variance (ANOVA).

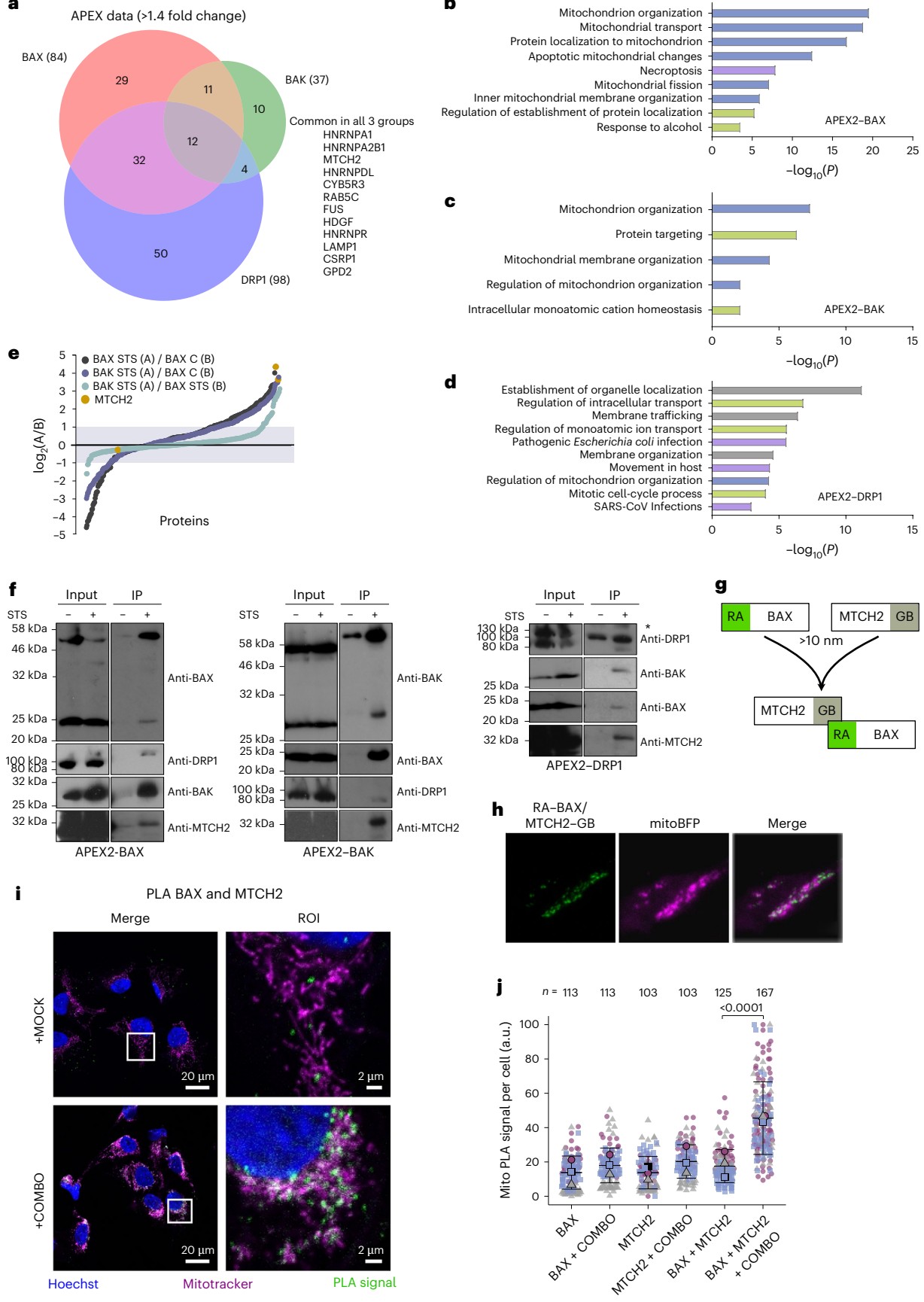

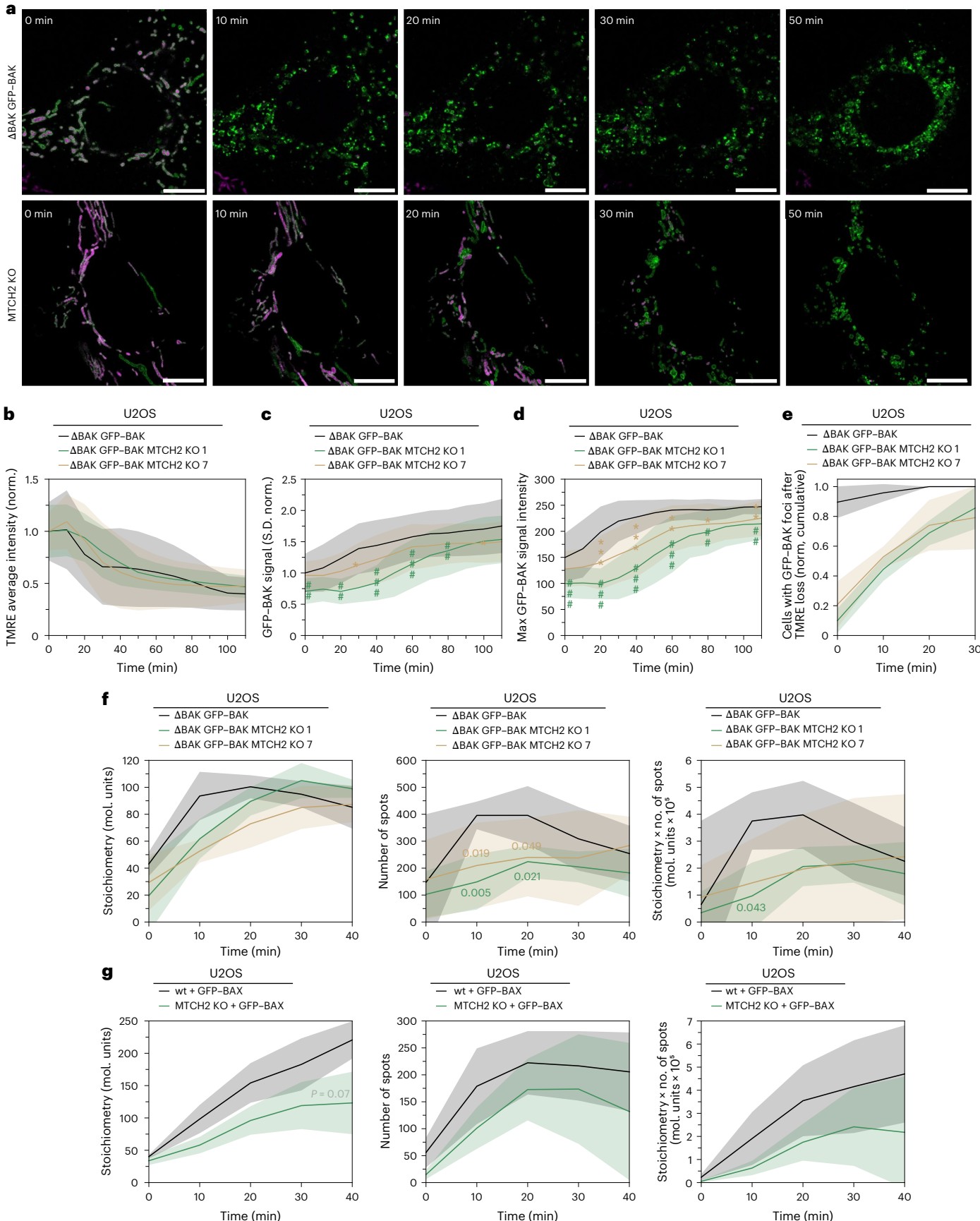

**Fig. 3 | MTCH2 depletion inhibits BAX and BAK high-order oligomerization in apoptosis and uncouples it from MOMP. a**, Representative images of the kinetics of mitochondrial potential loss as a proxy for MOMP and BAK oligomerization into apoptotic foci in U2OS ΔBAK cells expressing GFP–BAK (top) and MTCH2-KO U2OS ΔBAK cells expressing GFP–BAK (bottom) treated with 1 μM COMBO and 10 μM QVDA for 120 min. GFP–BAK is shown in green and TMRE is shown in magenta. Scale bar, 10 μm. **b–e**, Quantification of mitochondrial potential loss and BAK oligomerization in U2OS ΔBAK cells expressing GFP–BAK and two single clones of MTCH2-KO U2OS ΔBAK cells expressing GFP–BAK. **b**, Average TMRE signal over time normalized to each cell type. **c,d**, Accumulation of BAK into discrete foci quantified as the s.d. of the GFP signal (**c**) or GFP maximum signal (**d**) over time. Values are normalized to wt cells. **e**, Cumulative quantification of the number of cells presenting GFP–BAK foci at different times after TMRE loss. Experiments in **d,e** were performed with >20 cells from $n = 2–3$ independent experiments. Solid lines connect mean values at different time points. Colored areas correspond to the s.d. **f**, Left, mean stoichiometry of GFP–BAK foci at different time points in U2OS ΔBAK cells

expressing GFP–BAK and MTCH2-KO U2OS ΔBAK cells expressing GFP–BAK treated with 1 μM COMBO and 10 μM QVD. Middle, number of GFP–BAK foci (spots) over time. Right, amount of oligomerized GFP–BAK over time (calculated as monomers per focus multiplied by the number of foci). Solid lines connect mean values at different time points. Colored areas correspond to the s.d. **g**, Left, mean foci stoichiometry at different time points in wt and MTCH2-KO U2OS cells expressing full-length GFP–BAX treated with 1 μM COMBO and 10 μM QVD. Middle, number of GFP–BAX foci (spots) over time. Right, amount of oligomerized GFP–BAX over time (calculated as monomers per focus multiplied by the number of foci). Solid lines connect mean values at different time points. Colored areas correspond to the s.d. $P$ values were calculated comparing wt and MTCH2-KO cells at the indicated time points in **c** and **d**, using an unpaired, two-sided, Welch-corrected $t$-test. $*P < 0.05$, $**P < 0.01$ and $***P < 0.001$ comparing wt and MTCH2-KO7 at a concrete time point. $^{\#}P < 0.05$, $^{\#\#}P < 0.01$ and $^{\#\#\#}P < 0.001$ comparing wt and MTCH2-KO1 at a concrete time point. A table with the exact $P$ values can be found in Supplementary Table 3.

by the absence of MTCH2. This indicates that MTCH2 impacts BAX and BAK oligomerization downstream of their activation by BH3 mimetics.

### Reduced BAX and BAK assembly under MTCH2 depletion can be rescued by LPA

Together with cardiolipin (CL), MTCH2 mediates the recruitment of tBID, the caspase-8-cleaved version of the proapoptotic protein BID that activates BAX and BAK, to mitochondria during apoptosis[25,29,30]. Yet our findings regarding the proximity between MTCH2 and BAX/BAK and its effect on their oligomerization were performed under conditions in which BID remained inactivated (Supplementary Fig. 2b). To further discard a potential role of BID, we knocked out BID in U2OS ΔBAK cells stably expressing GFP–BAK, deficient or not in MTCH2, and confirmed that BID deficiency did not impact the delay in BAK focus formation and oligomerization kinetics upon MTCH2 KO or their rescue by MTCH2 reconstitution (Supplementary Fig. 2c–g). Together, these results show that MTCH2 also has a direct effect on BAX and BAK function independent of its function on tBID recruitment.

MTCH2 has been proposed to act as a mitochondrial insertase facilitating the correct insertion of tail anchor (TA)-containing proteins at the MOM[23]. As both BAX and BAK bind to the MOM through their TA, we investigated whether the insertase activity of MTCH2 was responsible for promoting BAX and BAK assembly during apoptosis. To test this hypothesis, we measured the impact of MTCH2 KO on apoptotic focus formation using BAX and BAK proteins lacking their TA domain, GFP–BAKΔC and GFP–BAXΔC. GFP–BAKΔC did not form detectable foci in either wt or MTCH2-KO cells, in line with the requirement of the TA for mitochondrial targeting[39–41]. We detected binding of BAXΔC to mitochondria, which may be mediated by endogenous

BAX and BAK or by membrane insertion of its pore-forming helices 5 and 6 (refs. 14,42). GFP–BAXΔC foci, characterized by lower number and stoichiometry compared to the full-length GFP–BAX constructs, could be detected in both cell lines during apoptosis (Fig. 4a–c). This is consistent with previous studies in which the BAX TA was linked to BAX assembly kinetics and pore growth[43]. However, only a minor additional impact was observed when both the TA of BAX and MTCH2 were absent (Fig. 4c). As an alternative approach, we generated the D189R mutant reported to abolish MTCH2 insertase activity[23] and cotransfected it or wt MTCH2 with GFP–BAX in MTCH2-KO U2OS cells (Fig. 4d). Reintroduction of wt MTCH2–mCherry restored GFP–BAX assembly kinetics in MTCH2-KO U2OS cells to the levels of the wt cells (Figs. 3g and 4e), supporting the MTCH2 specificity of our observations. However, MTCH2-R189D expression rescued GFP–BAX assembly similarly to the wt protein, except for a slight reduction in the number of detected foci (Fig. 4e).

Furthermore, we found that MTCH1, a paralog of MTCH2 with insertase activity[23], had no effect on GFP–BAK foci growth in ΔBAK U2OS cells stably expressing GFP–BAK (Fig. 4f,g). We also analyzed the mitochondrial targeting of BAK in untreated wt and MTCH2-KO mouse embryonic fibroblasts (MEFs), as well as HeLa and U2OS cells, and found that BAK primarily located to crude mitochondria fractions irrespective of MTCH2 levels (Supplementary Fig. 3a–c). Altogether, these data indicate that, while the insertase activity of MTCH2 might have a role, it is not essential for the effect of MTCH2 on the regulation of BAX and BAK high-order oligomers.

MTCH2 is also a regulator of lipid metabolism and has previously been associated with obesity[24,27,31,32]. As the lipid composition of mitochondria impacts the assembly of the apoptotic pore[44],

**Fig. 4 | The effect of MTCH2 on BAX and BAK oligomerization is independent of its protein insertase activity. a**, Representative images of the kinetics of mitochondrial potential loss as a proxy for MOMP and BAX oligomerization into apoptotic foci in wt U2OS cells expressing GFP–BAX (top) treated with 1 μM COMBO and 10 μM QVDA or GFP–BAXΔC (bottom) treated with 5 μM COMBO and 10 μM QVDA for 90–120 min. GFP–BAX and GFP–BAXΔC are shown in green and TMRE is shown in magenta. Scale bar, 10 μm. Images are representative of $n = 3$ independent experiments. **b**, Structure of inactive monomeric BAX (PDB 1F16) with the TA domain labeled in magenta. The structure was rendered using PyMOL. **c**, Left, mean focus stoichiometry at different time points in wt and MTCH2-KO U2OS cells expressing GFP–BAXΔC treated with 5 μM COMBO and 10 μM QVD. Middle, number of GFP–BAXΔC foci (spots) over time. Right, amount of oligomerized GFP–BAXΔC over time (calculated as monomers per focus multiplied by the number of foci). Solid lines connect mean values at different time points. Colored areas correspond to the s.d. $*P < 0.05$, comparing wt and MTCH2-KO cells at the indicated time point, calculated using an unpaired, two-sided, Welch-corrected $t$-test. **d**, Putative structure of MTCH2

(AF-Q9Y6C9-F1-model_v4) with D189 labeled in red. The structure was rendered using PyMOL. **e**, Left, mean focus stoichiometry at different time points in wt and MTCH2-KO U2OS cells expressing GFP–BAX and mCherry–MTCH2 or mCherry–MTCH2-D189R treated with 1 μM COMBO and 10 μM QVD. Middle, number of GFP–BAX foci (spots) over time. Right, amount of oligomerized GFP–BAX over time (calculated as monomers per focus multiplied the number of foci). Solid lines connect mean values at different time points. Colored areas correspond to the s.d. $*P < 0.05$, comparing wt and MTCH2-KO cells at the indicated time point, calculated using an unpaired, two-sided, Welch-corrected $t$-test. **f**, Representative WB of MTCH1 and MTCH2 expression levels in U2OS ΔBAK cells expressing GFP–BAK from $n = 2$ independent replicates. **g**, Left, mean stoichiometry of GFP–BAK foci at different time points in U2OS ΔBAK cells expressing GFP–BAK and MTCH1-KO U2OS ΔBAK cells expressing GFP–BAK treated with 1 μM COMBO and 10 μM QVD. Middle, number of GFP–BAK foci (spots) over time. Right, amount of oligomerized GFP–BAK over time (calculated as monomers per focus multiplied by the number of foci). Solid lines connect mean values at different time points. Colored areas correspond to the s.d.

we hypothesized that MTCH2 might promote BAX and BAK assembly through regulation of the mitochondrial lipid content. To test this, we isolated mitochondria form wt and MTCH2-KO MEFs and extracted their lipids, which we quantified by MS. As shown in Fig. 5a,b, we detected a decrease in several major phospholipid classes in MTCH2-KO cells compared to wt cells, of which CL was most dramatically reduced. In line with alterations of lipid metabolism, we also observed accumulation of lipid droplets in MTCH2-deficient cells (Supplementary Fig. 4), in agreement with a previous study[32].

Recent studies reported that MTCH2 cooperates with MFN2 to promote mitochondrial fusion dependent on LPA as a lipogenesis intermediate[25,26]. To evaluate whether a similar mechanism could take place under our experimental conditions, we exogenously added LPA to ΔBAK U2OS cells lacking MTCH2 or not and expressing GFP–BAK and measured apoptotic foci formation. Remarkably, addition of LPA partially rescued the uncoupling between TMRE loss and apoptotic focus formation observed in MTCH2-KO cells (Fig. 5c), as well as in the absence of BID (Supplementary Fig. 2h–k). To control that LPA

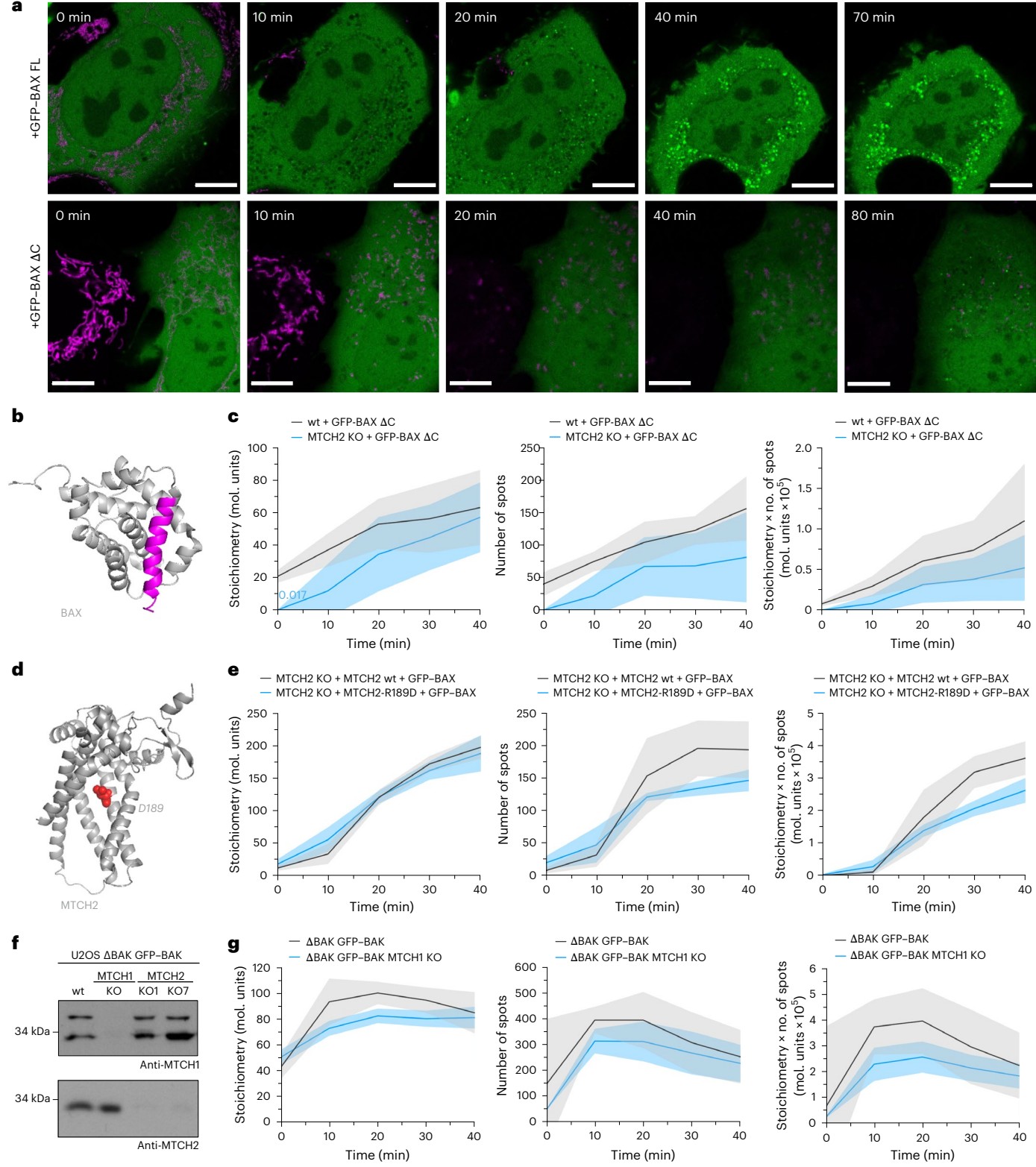

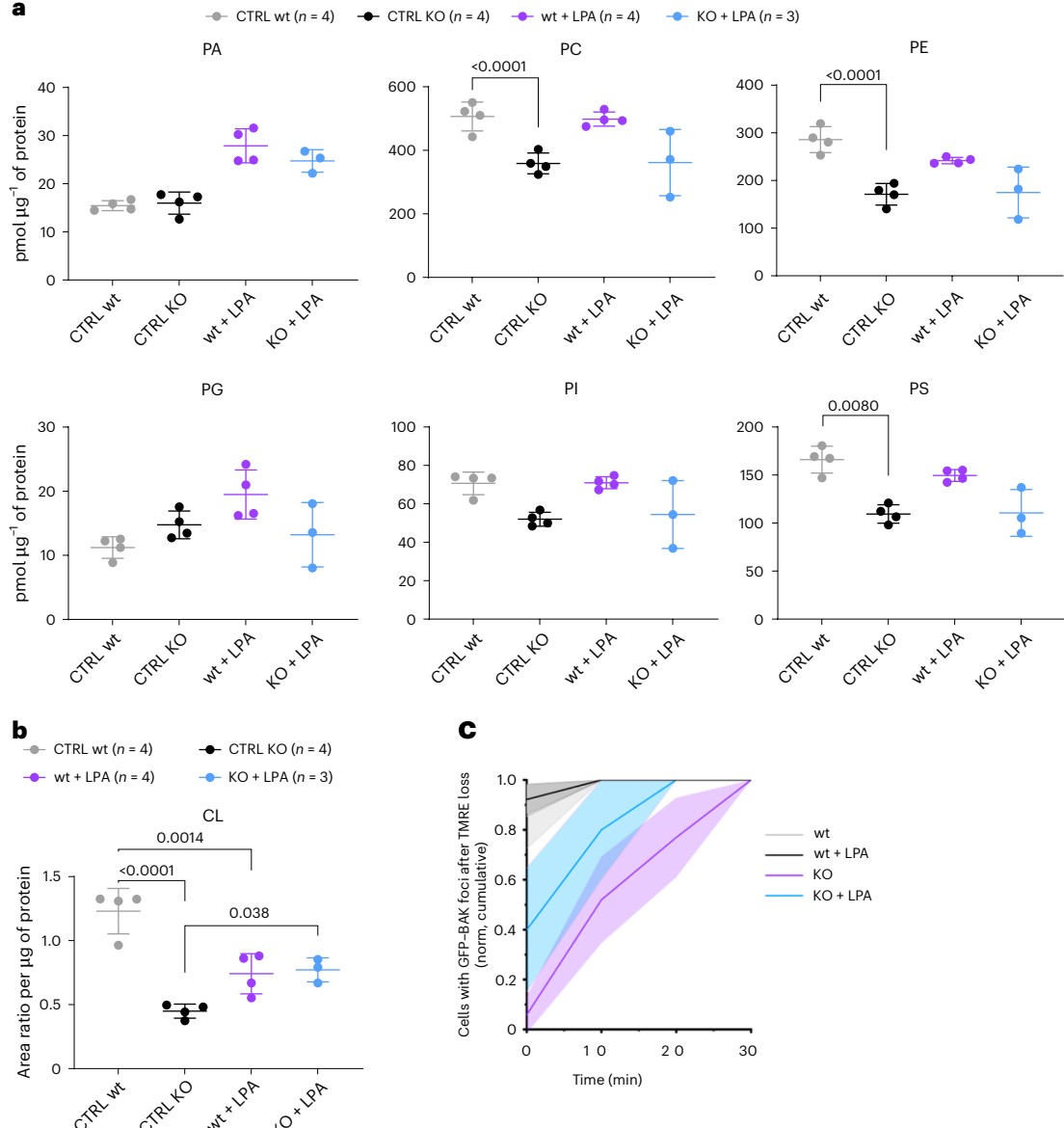

**Fig. 5 | BAX and BAK oligomerization delay under MTCH2 depletion can be rescued by treatment with LPA. a,b,** Phospholipid profile of mitochondria isolated from wt and MTCH2-KO U2OS ΔBAK cells expressing GFP–BAK and treated or not with LPA. Data are presented as the mean ± s.d. (n = 3 (KO + LPA) and 4 (others) from independent experiments). **a,** Analyses of phospholipid species PC, PE, PI, PS, PG and PA were performed in positive ion mode. P values were calculated using an unpaired two-way ANOVA corrected for multiple comparisons using Tukey's multiple-comparison test. **b,** Analysis of CL was performed in negative ion mode. P values were calculated using an unpaired,

two-way t-test. **c,** Cumulative quantification of number of cells presenting GFP–BAK foci at different times after TMRE loss. U2OS ΔBAK cells expressing GFP–BAK and MTCH2-KO U2OS ΔBAK cells expressing GFP–BAK treated with 1 μM COMBO and 10 μM QVDA. Cells were incubated in medium with delipidated serum, in the presence or absence of 20 μM LPA for 18 h, before Lightning microscopic analysis (>20 cells per group from n = 3 independent experiments were analyzed and presented in the graph). Solid lines connect mean values at different time points. Colored areas correspond to the s.d.

treatment reached the mitochondria, we confirmed the increase in levels of phosphatidic acid (PA) in the mitochondria by MS (Fig. 5a,b). Interestingly, LPA treatment also increased CL content in mitochondria. Together, these results suggest that it is not the protein insertase activity of MTCH2 but its function associated with lipid metabolism that may promote the high-order assembly of BAX and BAK during apoptosis.

## MTCH2 favors mtDNA release and cGAS–STING activation during apoptosis

Prompted by the effect of MTCH2 on BAX and BAK high-order assembly, we evaluated the impact of MTCH2 on mtDNA release during apoptosis. We induced apoptosis in wt, BAX/BAK-DKO (double KO), MTCH2-KO

and MTCH2-KO MEFs in which we reintroduced MTCH2, hereafter MTCH2 knock-in (KI), and quantified by qPCR the mtDNA content in the cytosolic fractions at different time points after COMBO treatment in presence of caspase inhibitors. Consistent with MTCH2 favoring BAX and BAK assembly, apoptotic pore growth and mtDNA release, wt MEFs presented faster mtDNA release kinetics than MTCH2-KO MEFs (Fig. 6a). Importantly, MTCH2-KI cells partially rescued the phenotype, supporting the specific effect of MTCH2 on apoptotic pore growth. We also cotransfected wt and MTCH2-KO HeLa cells with SMAC–GFP, an IMS protein released upon MOMP, with TFAM–RED, a transcription factor associated with mtDNA, and with Halo–TOM, a marker for the MOM. This allowed us to simultaneously visualize MOMP, the subcellular

localization of mtDNA and the MOM, showing faster release kinetics of TFAM−RED and, thus, mtDNA in HeLa wt cells compared to MTCH2-KO cells (Fig. 6b,c).

Mitochondrial crista shape has been associated with mtDNA release[45]. Under healthy conditions, both wt and MTCH2-KO cells showed similar crista morphology; however, during apoptosis, crista remodeling associated with apoptosis (crista swelling and enlargement of the intracrista distance) was reduced upon MTCH2 depletion (Fig. 6d,e and Supplementary Fig. 5a–e). We also analyzed the effect of MTCH2 on mitochondria–endoplasmic reticulum distance with the BRET-based biosensor MERLIN[46,47]. Consistent with previous studies, the BRET signal indicative of proximity between the organelles increased during apoptosis in both HeLa and U2OS cells. However, no significant differences were observed in MTCH2-KO cells during apoptosis (Supplementary Fig. 5f).

Once in the cytosol, mtDNA can activate the cGAS−STING pathway, causing STING degradation and TBK1 phosphorylation, in the absence of caspase activity[48]. We thus induced apoptosis followed by subcellular fractionation and analyzed the cytosolic fractions for cyt c and TFAM as indicators of MOMP and mtDNA release, respectively, and the membrane fractions were analyzed for STING degradation and TBK1 phosphorylation (Fig. 6f,g and Supplementary Fig. 5g,h). Interestingly, wt, MTCH2-KO and MTCH2-KI MEFs released cyt c with similar kinetics. The total amount of released cyt c was only partial in MTCH2-KO cells. In the case of TFAM release, while wt and MTCH2-KI cells exhibited both similar kinetics and intensities, MTCH2-KO cells showed delayed and reduced TFAM release. Consistently, STING degradation was evident in both wt and MTCH2-KI MEF cells but not in MTCH2-KO cells and TBK1 phosphorylation was faster in wt cells than in MTCH2-KO cells (Fig. 6g). Altogether, these data indicate that MTCH2 deficiency results in reduced mtDNA release and subsequent cGAS−STING pathway activation under low caspase activity during apoptosis, accompanied by maintenance of mitochondrial cristae ultrastructure.

### MTCH2 contributes to sublethal apoptosis signaling during *H. pylori* infection

As MTCH2-KO cells still undergo apoptosis but present reduced BAX and BAK oligomerization and apoptotic pore growth, we reasoned that MTCH2 may impact cell death sensitivity at lower concentrations of apoptosis triggers, as well as cellular responses to sublethal apoptosis signaling. Accordingly, we found that MTCH2 deficiency reduced cell death sensitivity (measured by DRAQ7 positivity) as we titrated down BH3 mimetics in U2OS and in HeLa cells (Fig. 7a and Supplementary Fig. 6a,b). Recent studies reported a role for the sublethal induction of MOMP in driving a drug-tolerant persister phenotype in cancer cells with implications for relapse of resistant tumors[49–51]. Using a similar procedure as in ref. 49, we found that MTCH2 depletion increased the number of surviving HeLa cells upon treatment with high doses of BH3 mimetics (Fig. 7b). Remarkably, the surviving MTCH2-KO cells also presented enhanced resistance to subsequent treatment with BH3 mimetics compared to surviving wt HeLa cells (Fig. 7c), indicating a role of MTCH2 in strengthening drug tolerance.

We also analyzed the role of MTCH2 using a physiologically relevant system of BAX and BAK activation in which sublethal MOMP has a role. For this, we infected wt and MTCH2-KO AGS gastric carcinoma cells with *H. pylori* (Supplementary Fig. 6c), a Gram-negative bacterium that can cause peptic ulcers and has previously been shown to induce sublethal BAX/BAK-dependent MOMP and caspase 3 activation, DNA damage and genomic instability[52]. Remarkably, upon *H. pylori* infection, MTCH2-deficient cells showed less phosphorylated γH2AX, indicative of reduced DNA damage, compared to wt cells (Fig. 7d,e). Similar results were obtained in HeLa cells (Supplementary Fig. 6d,e).

Collectively, these data suggest that MTCH2 contributes to sensitizing apoptotic signaling under lethal and sublethal stimuli, as well as DNA damage upon bacterial infection.

## Discussion

The opening of the apoptotic pore by BAX and BAK is integral to the execution of the intrinsic apoptotic pathway and their mode of action remains unclear. Here, we mapped the protein environment of the apoptotic BAX/BAK pore using in situ proximity labeling, which uncovers MTCH2 as a key regulator of pore assembly, growth and content release.

We also analyzed DRP1, which colocalizes with BAX and BAK and regulates their function[35]. While BAX and BAK showed highly similar protein environments, consistent with their coassembly into the same individual nanostructures[14], DRP1 presented a number of distinct enriched proteins. This suggests that it assembles close by rather than as a core part of the apoptotic pore or that it additionally associates with other cellular environments during apoptosis. Consistently, DRP1 was highly enriched in both APEX2−BAX and APEX2−BAK apoptotic samples.

Comparing the labeling of APEX2−BAX and APEX2−BAK in healthy and apoptotic cells identified several RNA metabolism-related proteins enriched during apoptosis, possibly reflecting the release of mtRNA from mitochondria[53]. Among apoptosis-linked mitochondrial proteins, we identified MIM proteins, such as ATAD3, YME1L1 and SLC25A24, suggesting that the mitochondrial reorganization during apoptosis brings the MIM in close proximity to the MOM at the apoptotic pore. This is consistent with earlier reports of cristae remodeling and MIM extrusion through the apoptotic MOM pore[6,7]. Whether these MIM proteins actively regulate the pore remains to be determined.

MTCH2 clearly stood out as a top enriched protein in the vicinity of BAX, BAK and DRP1 during apoptosis. The function and molecular mechanism of MTCH2 remains poorly understood. Its role in obesity and lipid metabolism is exerted through yet unknown mechanisms, which have been associated with lipid fluxes[25,26]. Although MTCH2 has a role together with CL in the recruitment to tBID to mitochondria during

**Fig. 6 | MTCH2 promotes mitochondrial crista redistribution and mtDNA release during apoptosis. a**, qPCR data ($\Delta C_t$ values) of DNA amounts extracted from cytosols of apoptotic MEF cells with four different primers that recognize mtDNA (Dloop, S16, non-nuMT and cytochrome b): wt MEFs (black), BAX/BAK-DKO MEFs (gray), MTCH2-KO MEFs (magenta) and MTCH2-KI MEFs (blue). Cytosols were collected at 0, 1, 2 and 3 h after treatment with 10 µM COMBO and 20 µM QVD. Left, dashed lines represent a trend in mtDNA liberation over time from independent samples in each technical replicate. The bold solid line represents the average of dashed lines. Each data point represents the mean of three technical replicates (*n* = 3 independent experiments). Right, quantification of the area under the dashed curve (AUC) for the different cell types and primer combinations. Error bars correspond to the s.d. **b**, Representative confocal images of wt and MTCH2-KO HeLa cells expressing TFAM−RED (red), SMAC−GFP (green) and Halo−TOM (gray) and treated with 10 µM COMBO and 20 µM QVD. Scale bars, 10 µm and 1 µm (zoomed-in areas). **c**, Quantification of cytosolic TFAM−RED in wt and MTCH2-KO HeLa cells treated as in **b** at indicated time points. The total number of cells considered from *n* = 3 independent experiments is shown in brackets. **d**, Representative EM images of mitochondrial cristae from U2OS ΔBAK cells expressing GFP−BAK and MTCH2-KO U2OS ΔBAK clones 5 and 8 expressing GFP−BAK treated with 5 µM COMBO and 20 µM QVD for 3 h. Scale bar, 500 nm. **e**, Quantification of cristae morphology of samples in **d**, classified into no cristae, abnormal cristae (swollen, abnormal distribution), normal (classical elongated cristae with uniform distribution) and mixed phenotype (*n* = 4–6 cells). The total number of mitochondria considered in this assay is shown in brackets. **f,g**, Representative WB of cGAS−STING pathway activation in wt and MTCH2-KO clone 5 MEFs and MTCH2-KI clone 5 MEFs at indicated time points after treatment with 10 µM COMBO and 20 µM QVD. In **f**, cytosolic fractions were analyzed for the presence of cyt c and TFAM. In **g**, pellets were checked for TBK1 phosphorylation and STING expression levels. P.S., Ponceau staining. Quantification of the blots is presented in Supplementary Fig. 8.

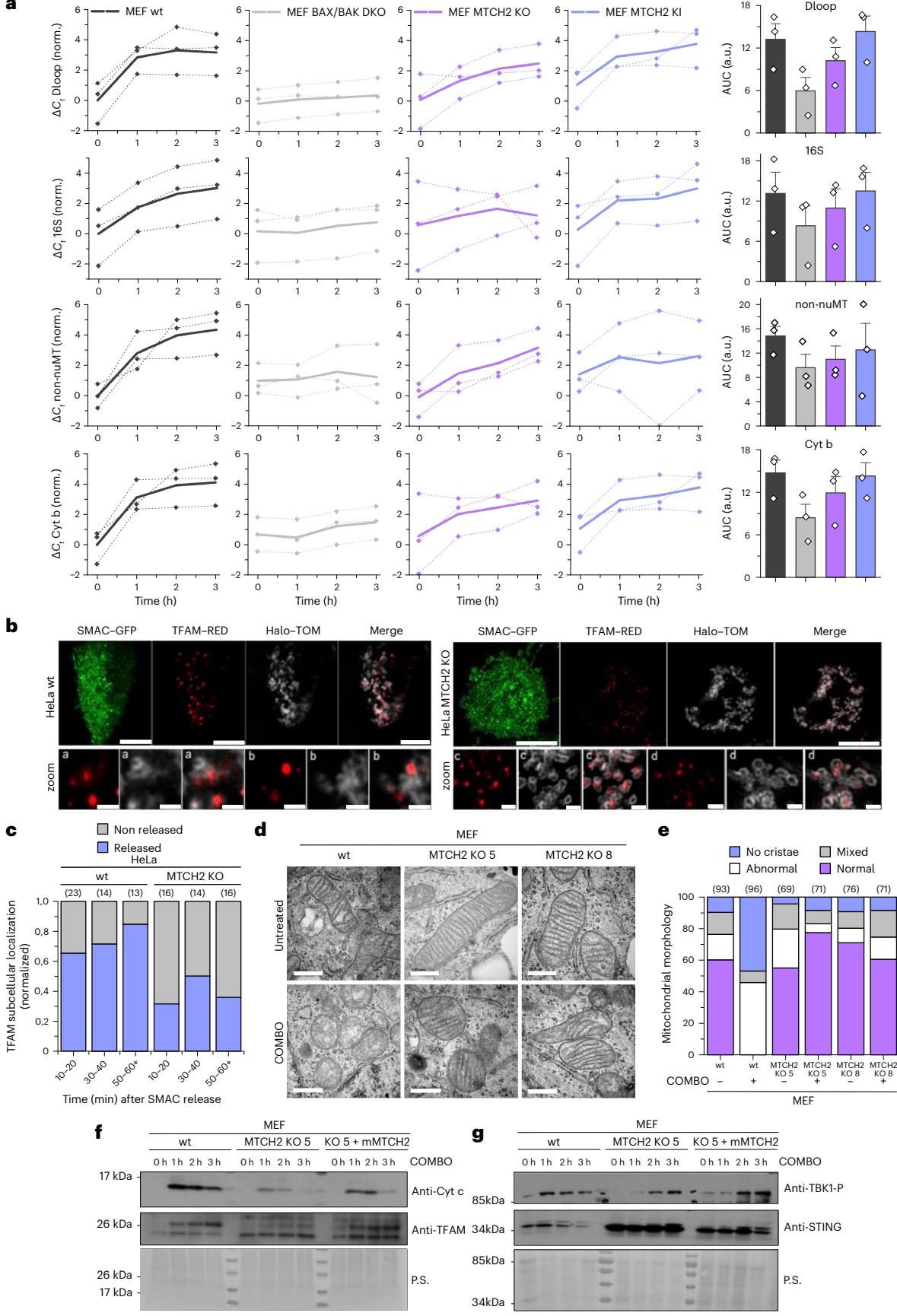

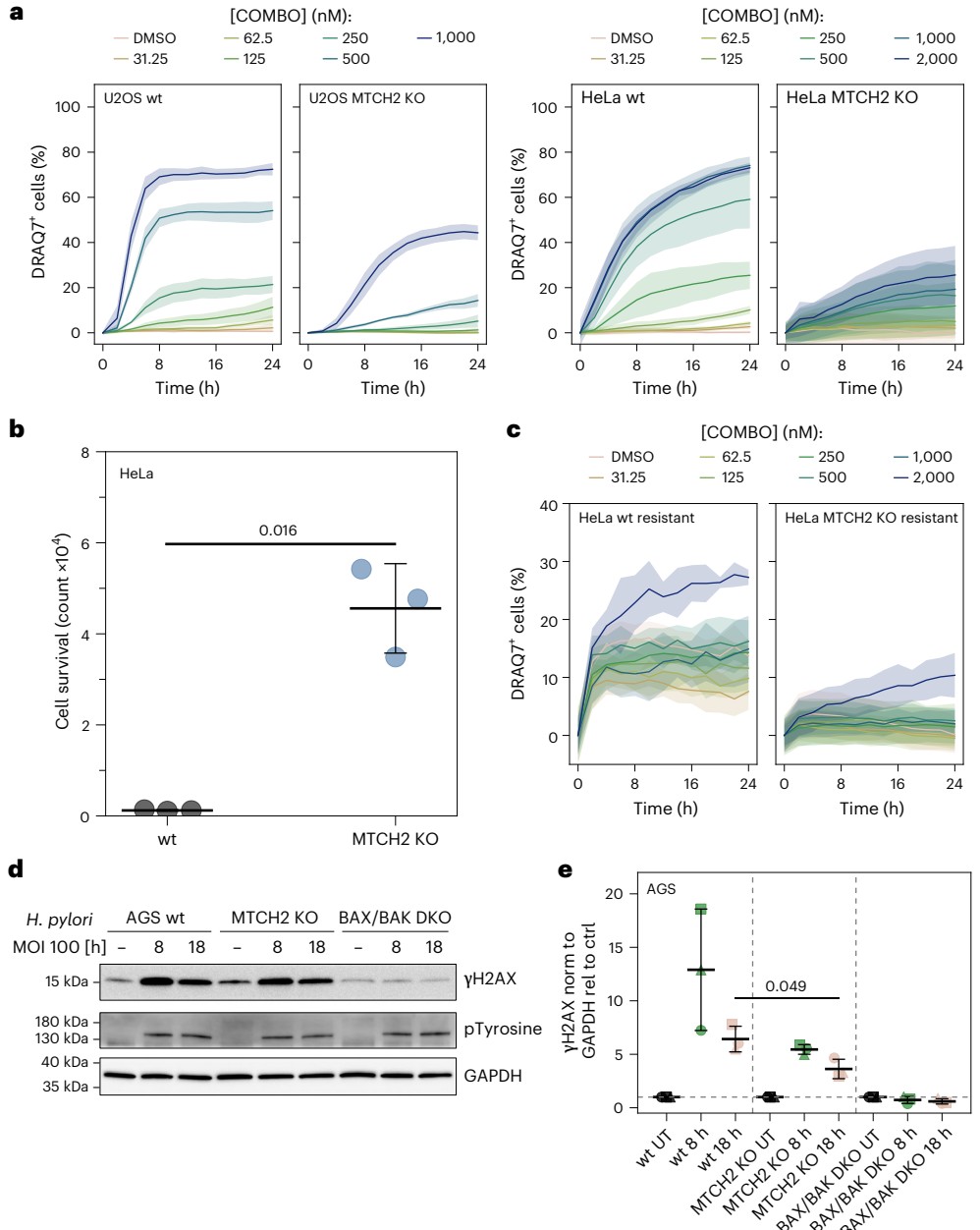

**Fig. 7 | MTCH2 contributes to sublethal apoptotic signaling during bacterial infection. a**, Quantification of cell death kinetics measured as the percentage of DRAQ7-positive cells in wt and MTCH2-KO HeLa and U2OS cells treated with the indicated concentrations of COMBO. Solid lines represent mean of three independent experiments. Colored areas correspond to the s.d. **b**, Quantification of drug-tolerant cell survival in wt and MTCH2-KO HeLa cells after treating them with COMBO 10 μM for 24 h and letting them recover for 7 days. Data are presented as the mean ± s.d. Individual points represent technical replicates from three independent experiments. The *P* value was determined using an unpaired, two-sided *t*-test (wt versus MTCH2-KO cells). **c**, Quantification of cell death kinetics measured as the percentage of DRAQ7-positive cells in wt and MTCH2-KO drug-tolerant HeLa cells treated with the indicated concentrations of

COMBO. Resistant cells were selected by treating them with COMBO 2 μM for 24 h and letting them recover for 1 day. Solid lines represent the mean of three independent experiments. Colored areas correspond to the s.d. **d**, Representative WB of γH2AX and tyrosine phosphorylation in cellular extracts of wt, BAX/BAK-DKO and MTCH2-KO AGS cells infected or not (−) with *H. pylori* (*HP*). GAPDH was used as a loading control (*n* = 3 independent experiments). **e**, Quantification of data in **d**. γH2AX levels upon *H. pylori* infection in the indicated samples were normalized to the total protein concentration and to the control of each specific cell line. Data are presented as the mean ± s.d. Individual points represent technical replicates from three independent experiments. The *P* value was determined using an unpaired, two-sided *t*-test (wt versus MTCH2-KO cells).

apoptosis[29,30], our results uncover a tBID-independent role of MTCH2 in the regulation of BAX and BAK assembly into apoptotic pores.

Mechanistically, MTCH2 does not seem to require its insertase activity to tune BAX and BAK oligomerization. This idea is supported by the fact that MTCH1, an analog of MTCH2 retaining TA insertase activity, fails to modify BAK self-assembly kinetics. Cells expressing a MTCH2 mutant lacking insertase activity exhibit no differences in GFP–BAX stoichiometry.

Furthermore, in the absence of the TA domain, BAX displays diminished but comparable assembly in both wt and MTCH2-KO cells.

Our results suggest that it is instead MTCH2 lipid metabolism-related activity that impacts the assembly of BAX and BAK during apoptosis. We found that MTCH2 depletion altered mitochondrial lipid composition and caused a decrease in several major classes, including CL. This is in line with a role of MTCH2 in cellular metabolism leading to

changes in membrane lipids, which might be linked to its lipid scramblase activity. Although the structural mechanism for the lipid-related activity of MTCH2 is still unknown, it is tempting to speculate that it could regulate the lipid content of mitochondria, thereby providing the adequate membrane environment for apoptotic pore growth at the MOM. Of note, such model may not require direct interaction between MTCH2 and BAX/BAK. In line with this notion, BAX and BAK are known to establish protein–lipid pores of tunable size[12] and the presence of polyunsaturated lipids also favors apoptotic pore growth[44]. Remarkably, the recovery of BAK self-assembly kinetics by treatment with LPA suggests that MTCH2 deficiency might affect mitochondria lipid fluxes and, thus, their membrane composition, in a way that can be bypassed by doping mitochondria with this soluble lipid precursor. While this hypothesis is in good agreement with previous studies[25,26], we cannot discard an indirect effect of LPA on MOMP sensitivity, although reports in the literature suggest a rather antiapoptotic effect[54–56]. Furthermore, we cannot discount an effect on the recruitment of BH3-only proteins other than BID, although, unlike tBID, they have been reported to insert in membranes through their hydrophobic C-terminal anchors[57–61].

We also provide evidence for the functional relevance of this alternative role of MTCH2. We observed reduced release of cyt c, which could be because of the reduced crista remodeling that is still sufficient to induce caspase activation and cell death. Of note, a notable fraction of cyt c is trapped inside cristae and their rearrangement is necessary for complete cyt c release[62–64]. In contrast, we found that cells lacking MTCH2 presented reduced mtDNA release and cGAS–STING pathway activation during apoptosis. These findings suggest that the promotion of apoptotic pore growth by MTCH2 is important for assembling larger pores that allow the release of mtDNA. Under low caspase activity, this then impacts downstream activation of cGAS–STING and inflammatory signaling[5,8,65,66]. Our results indicate that MTCH2 also promotes BAX and BAK function under conditions of sublethal or minority MOMP in the context of bacterial infection and it may have a role in *H. pylori*-induced genomic instability. These activities could be related to differential pore size formation or pore formation kinetics, which will require further studies.

In summary, these findings establish MTCH2 as an important regulator of the high-order assembly and growth of the apoptotic pore, which likely exerts this function by tuning the mitochondrial membrane composition. Our findings advance the current understanding of the molecular mechanisms governing the apoptotic pore and the contribution of mitochondria to regulating the pore assembly. They also have consequences for inflammatory signaling and sublethal apoptosis upon bacterial infection, thus opening up new possibilities for specifically targeting the growth of the apoptotic pore in therapeutic strategies.

## Online content

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

## Methods

### Plasmids and antibodies

All plasmids and antibodies used in this study are listed in Supplementary Table 1.

### Cell culture

All cells used in this study were cultured in high-glucose DMEM (Gibco) supplemented with 10% FBS and 1% penicillin–streptomycin (Invitrogen). Cells were transfected at 70–80% confluence. All cell lines used in this study were subjected to regular *Mycoplasma* testing.

### Cell line generation

HeLa wt cells were bought from the American Type Culture Collection (CCL2), U2OS NUP96–mEGFP cells were obtained from the CLS Cell Lines Service (300174) and AGS wt cells were bought from Sigma-Aldrich (89090402-1VL). MTCH2-KO U2OS, MTCH2-KO U2OS ΔBAK GFP–BAK, BID-KO U2OS ΔBAK GFP–BAK, BID-KO MTCH2-KO U2OS ΔBAK GFP–BAK MTCH2-KO HeLa and MTCH1-KO U2OS ΔBAK GFP–BAK cells were generated in their respective wt or parental cell types by CRISPR–Cas9. Guide RNAs were designed to target the N-terminal region of the genes using the UCSC Genome Browser and selected with CRISPOR on the basis of high MIT specificity scores. CRISPR–Cas9 knockouts were generated by transfecting the pU6-(BbsI)_CBh-Cas9-T2A-BFP vector with selected gRNA (Supplementary Table 1), as described previously[67,68]. Here, $5 \times 10^5$ cells were seeded in a six-well plate and 500 ng of CRISPR constructs were transfected with 1.5 µl of PEI according to the manufacturer's instructions. Then, 24 h after transfection, cells were sorted for BFP-positive cells. MTCH2-KO U2OS BFP-positive cells were collected in bulk, whereas, in U2OS ΔBAK GFP–BAK, BID-KO, MTCH2-KO, MTCH2/BID-DKO and MTCH1-KO cells, individual clones were isolated. We produced bulk and single-clone MTCH2-KO HeLa cells. Success of the knockout was validated by WB analysis.

For reconstitution of mouse MTCH2 in MTCH2-KO MEFs, we cloned the open reading frame (ORF) from mouse MTCH2 (GenEZ ORF clone: Mtch2_OMu14693D_pcDNA3.1+/C-(K)-DYK) into Gateway pENTR 11 dual as described previously[69]. Next, the mouse MTCH2 sequence was then introduced into pLenti CMV Puro DEST (w118-1) using Gateway LR Clonase II enzyme mix from Invitrogen (Germany). For lentivirus production, HEK-293T cells were cotransfected with the lentiviral vectors and the packaging vectors PMD2.G and PSPAX2. The viruses were isolated by centrifugation for 5 min at 500g and target cells (MTCH2-KO MEFs) were then infected with the virus using polybrene infection/transfection reagent from Sigma-Aldrich. MEFs expressing mMTCH2 were selected by treatment with 2.5 µg ml⁻¹ puromycin for 7 days (MTCH2-KI MEFs). The U2OS ΔBAK GFP–BAK cell line was generated by introducing GFP–BAK into U2OS ΔBAK cells with G418 selection (0.7 mg ml⁻¹) for 3 weeks and colony isolation. Success of the GFP–BAK reintroduction was validated by WB analysis.

### APEX2 experiments

Biotinylation experiments were performed as described previously[34]. HeLa cells were incubated for 3 h with 1 µM STS at 37 °C, 5% $CO_2$ to induce apoptosis or remained untreated. Next, the medium was changed to DMEM with 500 µM BP and cells were incubated for 30 min at cell culture conditions. $H_2O_2$ was then added to a final concentration of 1 mM in PBS for 1 min at RT to initiate the biotinylation reaction. Reactions were then quenched by washing three times with quenching solution (10 mM sodium ascorbate, 10 mM sodium azide and 5 mM Trolox in PBS).

**Immunostaining to detect APEX2 constructs and biotinylated proteins.** HeLa cells were seeded in 12-well plates onto glass coverslips and transfected with the APEX2 constructs together with mitoBFP plasmid for 16 h (Supplementary Table 1). After apoptosis and biotinylation reactions were induced as described above, cells were fixed with 4% PFA in PBS

for 15 min at room temperature and then washed three times with PBS. Next, cells were permeabilized with 0.25% Triton X-100 in PBS for 5 min at room temperature followed by three washes with PBS. The cells were blocked for 1 h at room temperature with 2% BSA in PBS. The samples were rocked in anti-FLAG 1:100 in 2% BSA for 1 h at 4 °C for immunostaining for the APEX2 fusion protein through its epitope tag. After primary antibody incubation, samples were washed four times with Tris-buffered saline and Tween-20 (TBST) and rocked in secondary antibody (1:500 anti-mouse AF488) and 1:10,000 streptavidin–AF555 in 2% BSA in PBS for 30 min at room temperature. Samples were washed four times with TBST and mounted in glass slides using ProLong Gold antifade. Images were acquired using a Zeiss LSM 710 ConfoCor3 microscope with an Apochromat ×63 (numerical aperture (NA): 1.4) oil-immersion objective.

### Proteomics experiments

**Sample preparation and chemical labeling.** To prepare the biotinylated proteome, two 15-cm dishes per condition at 70–80% confluency were used. Cells were seeded and transfected with combinations of the different APEX2 constructs into three independent experiments. After apoptosis and biotinylation reactions, cells were harvested and resuspended in RIPA buffer and incubated with prewashed streptavidin-coated magnetic beads for 1 h at 4 °C under continuous stirring. Beads were washed twice with RIPA buffer, once with 1 M potassium chloride, once with 0.1 M sodium carbonate, once with 2 M urea in 10 mM Tris-HCl pH 8.0, twice in RIPA buffer and three times with 20 mM ammonium bicarbonate pH 8.0.

**Dimethylation labeling, nanoLC–MS/MS analysis and data processing.** Proteins were digested on beads with trypsin, acidified peptides were purified using C18 stage tips and labeled with dimethyl 'light' $((CH_3)_2)$, dimethyl 'intermediate' $((CH_1D_2)_2)$, and dimethyl 'heavy' $((^{13}CD_3)_2)$. Peptide mixtures were analyzed on an Easy-nLC II coupled to an LTQ Orbitrap XL MS instrument, the peptides were eluted with a 127-min segmented gradient of 5–33–50–90% HPLC solvent B (80% acetonitrile in 0.1% formic acid) at a flow rate of 200 nl min⁻¹.

Precursor ions were acquired in the mass range from $m/z$ 300 to 2,000 in the Orbitrap mass analyzer at a resolution of 60,000. An accumulation target value of $10^6$ charges was set and the lock mass option was used for internal calibration. The ten most intense ions were sequentially isolated and fragmented in the linear ion trap using collision-induced dissociation (CID) at the ion accumulation target value of 5,000 and default CID settings. Sequenced precursor masses were excluded from further selection for 90 s.

Acquired MS spectra were processed with MaxQuant software package version 1.5.2.8 with integrated Andromeda search engine. A database search was performed against a target–decoy *Homo sapiens* database obtained from UniProt. Trypsin was defined as a protease with a maximum of two missed cleavages. Oxidation of methionine and N-terminal acetylation were specified as variable modifications, whereas carbamidomethylation on cysteine was set as a fixed modification. Light, intermediate and heavy dimethylation labeling on peptide N termini and lysine residues was defined. The initial maximum allowed mass tolerance was set to 4.5 ppm (for the survey scan) and 0.5 Da for CID fragment ions. Peptide, protein and modification site identifications were reported at a false discovery rate of 0.01, estimated by the target–decoy approach. Quantification of dimethyl-labeled peptides required at least two ratio counts. Perseus software (version 1.5.0.15) was used for calculation of the significance B (psigB) for each protein ratio with respect to the distance of the median of the distribution of all protein ratios, as well as its intensitiy. All proteins with psigB < 0.01 in a pairwise comparison were considered to be differentially regulated.

**Pathway enrichment analysis.** Proteins from three independent quantitative MS experiments (Fig. 1e) that fulfilled the first and second selection criteria were used to perform biological pathways enrichment analysis.

To this end, protein gene identifiers from protein list 1 (APEX2–BAX), protein list 2 (APEX2–BAK) and protein list 3 (APEX2–DRP1) were submitted to Metascape (http://metascape.org). Metascape is an open-access gene annotation and analysis resource, which uses a hypergeometric test and Benjamini–Hochberg P value correction algorithm to identify all ontology terms that contain a statistically greater number of genes in common with an input list than expected by chance. Metascape automatically clusters enriched terms into nonredundant groups and the most significant (lowest P value) term within each cluster is presented in a bar graph[70].

## WB

Cells were lysed in RIPA buffer (150 mM NaCl, 0.5% sodium deoxycholate, 1% NP-40, 0.1% SDS and 50 mM Tris pH 8.0) with protease and phosphatase inhibitors, incubated on ice for 30 min and centrifuged at 20,000g for 20 min. Supernatants were collected as cellular extracts. The protein concentration was determined by Bradford protein assay (Bio-Rad) according to the manufacturer's protocol. Equal amounts of protein (40–80 µg) were loaded on 4–12/15% Tris–Bis gels (Thermo Scientific) and transferred onto nitrocellulose membrane using the Turboblot (Bio-Rad). Blots were blocked with 5% milk or fatty acid free BSA in TBST and incubated overnight at 4 °C or 1 h at room temperature with the primary antibody (Supplementary Table 1), probed with secondary antibodies (Supplementary Table 1) and developed using enhanced chemiluminescence (BioTool).

When subcellular fractionation of mitochondria and cytosolic fractions was required, cells were lysed in permeabilization buffer (20 mM HEPES–KOH pH 7.5, 100 mM sucrose, 2.5 mM MgCl$_2$, 100 mM KCl, freshly added 0.025% (w/v) digitonin and protease and phosphatase inhibitor cocktail in PBS) for 10 min on ice and total cellular membranes were pelleted by centrifugation at 10,000g for 5 min at 4 C. After removing the supernatant (cytosolic fraction), the membranes were solubilized using RIPA buffer as described above.

WB validation of APEX2 biotinylation was achieved by obtaining whole-cell lysates were as described above and analyzing by WB with streptavidin-conjugated antibody. For visualization of the streptavidin immunoprecipitated proteins, after washing, a fraction of streptavidin beads were resuspended in loading buffer supplemented with 2 mM biotin and boiled for 5 min at 95 °C. Then, beads were discarded using a magnetic rack, such that the supernatant fraction contained biotinylated proteins, before analyzing by WB.

## PLA

U2OS cells were seeded in 12-well plates onto coverslips, incubated with 200 nM MitoTracker deep red (Thermo Fisher, M22426) at 37 °C for 30 min, washed with prewarmed medium and left untreated or treated for 100 min with 1 µM ABT-737 and S63845. PLA was performed according to the manufacturer's protocol (Duolink in situ fluorescence protocol, Duolink in situ detection reagents green, DUO92014). Cells were fixed with 4% paraformaldehyde in PBS for 10 min at room temperature, permeabilized with 0.01% saponin in 3% BSA for 30 min at room temperature and blocked with the Duolink blocking solution for 60 min at 37 °C. Then, cells were incubated overnight with mouse BAX 6A7 monoclonal antibody (1:100; Thermo Fisher, MA5-14003) and rabbit polyclonal MTCH2 antibody (1:250; Thermo Fisher, PA5-88873), as well as negative and positive controls to validate the assay (data not shown). Samples were washed, incubated with the respective PLA probes for 1 h at 37 °C, washed and ligated for 30 min at 37 °C. After washing, polymerase amplification was performed for 100 min at 37 °C. In the end, the nuclei were stained with Hoechst. Images were acquired by confocal microscope (LSM Meta 710, Zeiss) using a ×63 (NA: 1.5) oil-immersion objective at room temperature. Quantitative analysis was performed using Fiji software. The mean fluorescence intensity of the PLA signal was measured only at mitochondria (using the binary mask 'Otsu' to mark only the mitochondrial area) and normalized to the background signal.

## Analysis of GFP–BAX and GFP–BAK assembly kinetics

The role of MTCH2 in tuning the assembly kinetics of BAX and BAK was investigated using photon-counting confocal microscopy as described previously[14], with small modifications. Briefly, BAK assembly experiments were conducted in U2OS ΔBAK cells expressing GFP–BAK. For BAX, GFP–BAX or GFP–BAXΔC were transfected in wt and MTCH2-KO U2OS cells and, where indicated, cells were cotransfected with MTCH2–mCherry or MTCH2-R189D–mCherry (Supplementary Table 1). Transfection per well was performed with 100 ng of plasmid per eight-well glass-bottom slide (IBIDI) using a 1:3 ratio of Lipofectamine 2000 or PEI for 12 h. U2OS ΔBAK BID-KO MTCH2-KO cells expressing GFP–BAK were transfected with 75 ng of MTCH2 wt plus 75 ng of pCAGGs-BFP-empty for cell transfection detection. To assess the role of LPA in BAK assembly kinetics, cells were incubated for 18 h in high-glucose DMEM containing 10% delipidated FBS (Bio&SELL, FCS.LFS.0100) and penicillin–streptomycin, with or without 20 µM LPA.

Apoptosis was induced with 1 µM ABT-737 (Hölzel, HY-50907-10mM) and 1 µM S63845 (Hoelzel, 100741), referred to as COMBO, including 10 µM Q-VD-OPh (Hölzel, HY-12305-10mg), a pan-caspase inhibitor, for all conditions with the exception of experiments with GFP–BAXΔC, where we used 5 µM ABT-737, 5 µM S63845 and 10 µM Q-VD-OPh. Cells were labeled with 100 nM TMRE in DMEM and maintained at 37 °C and 5% CO$_2$ during imaging.

For U2OS ΔBAK cells expressing GFP–BAK, GFP–BAX or GFP–BAXΔC, confocal microscopy was performed on a TCS SP8 gSTED 3X inverted microscope (Leica Microsystems) equipped with a Plan-Apochromat ×63 (NA: 1.4) oil-immersion objective and a white-light laser (WLL). The pinhole size was adjusted to one Airy unit. Next, z stacks of eight images were collected with an interval of 0.3 µm for a total of 2 µm. A zoom factor of ×4 was used to obtain an image size of 46.13 × 46.13 µm at 1,024 × 1,024 pixels. Images were acquired with a pixel dwell time of 400 ns and a line average of 2. Single emitted photons were detected on a HyD detector. Kinetics analysis of GFP–BAX or GFP–BAK oligomerization was performed by photon-counting fluorescence microscopy acquiring z stacks every 10 min (for a time series of 40–60 min) after MOMP. MOMP was monitored by TMRE signal loss, a dye whose fluorescence is dependent on mitochondrial potential. We quantified GFP–BAK and GFP–BAX oligomerization kinetics by analyzing the s.d. of GFP–BAK signal over time, as well as quantifying the time of focus appearance referred to TMRE fluorescence loss. Mitochondrial depolarization was considered when the TMRE signal dropped >30% from the time zero value. We analyzed image intensities and signal distribution using ImageJ.

For U2OS ΔBAK BID-KO cells expressing GFP–BAK, confocal microscopy was performed on a Zeiss LSM 980 Airyscan 2 inverted microscope equipped with a Plan-Apochromat ×63 (NA: 1.4) oil-immersion objective and a WLL. The pinhole size was adjusted to one Airy unit. Next, z stacks of eight images were collected with an interval of 0.3 µm for a total of 2 µm. A zoom factor of ×3 was used to obtain an image size of 44.9 × 44.9 µm at 1,024 × 1,024 pixels. Images were acquired with a pixel dwell time of 380 ns and a line average of 2. Single emitted photons were detected on a BiG.2 GaAsP photomultiplier tube detector (Hamamatsu). MOMP monitoring and analysis of GFP–BAK oligomerization kinetics were performed as described above. Image intensities and signal distribution were analyzed using Python.

## Stoichiometry determination analysis by ratiometric approach

Stoichiometry determination analysis was performed as described previously[14], with small modifications. Briefly, single-particle detection and stoichiometry analysis were performed on maximum intensity z projections of images acquired by confocal microscopy in photon-counting mode using an in-house developed stoichiometry analysis software implemented in Python.

Bright spots were automatically detected using the difference of Gaussian method (Python skimage.feature.blobdog; http://scikit-image.org/docs/dev/api/skimage.feature.html) and subsequent thresholding. Selected particles were identified by a region of interest (ROI) of defined particle size (5 × 5 pixels) and a fitted two-dimensional Gaussian by which the foci could be localized. Background subtraction was performed by sampling an area around the ROI. Localized foci were then filtered on the basis of their distance to each other to avoid overlapping, a threshold to remove too dim foci and the full width at half maximum (σ) of Gaussian fit, as described previously[35]. Foci whose intensity maximum was in the outer frames of the $z$ stack were also discarded as the intensity information could be lost.

The stoichiometries of GFP–BAX and GFP–BAK were normalized using a cell line with NUP96 tagged with GFP (U2OS NUP96–mEGFP; CLS Cell Lines Service, 300174) as a standard, which has a known stoichiometry of 32 (ref. [71]). To obtain the stoichiometry $S$, the intensities of foci were then divided by the average value of NUP96 intensities obtained on the same day in the same cell line and multiplied by their known stoichiometry of 32, where $I_x$ denotes the corresponding intensities:

$$S_{\text{foci}} = 32 \frac{I_{\text{foci}}}{\langle I_{\text{NUP96}} \rangle}$$

### ddFP
ddFP analysis was conducted as described previously[35,37]. HeLa cells were seeded in eight-well chambers (IBIDI) and transfected with equimolar amounts of RA–BAX, MTCH2–GB and mitoBFP plasmids (Supplementary Table 1) using Lipofectamine 2000 as described above. Apoptosis was triggered with 1 µM STS in phenol-free DMEM for 3 h and cells were acquired using a Zeiss C-Apochromat ×40 (NA: 1.2) water-immersion objective equipped with an incubator at 37 °C and 5% $CO_2$.

### TFAM subcellular localization determination assay using Lightning microscopy
First, wt and MTCH2-KO HeLa cells were seeded in a grid-containing 35-mm dish (IBIDI) and transfected with 100 ng of SMAC(1–60)–GFP, TFAM–RED and Halo–TOM (Supplementary Table 1) 12 h before the experiment. Low-resolution fluorescence microscopy images were acquired to identify SMAC release kinetics using an Echo Revolve fluorescence microscope (RVL-100-M). Apoptosis was induced with 10 µM COMBO including 20 µM QVDA, supplemented with 300 nM Janelia Fluor 633 for 60–90 min. Next, cells were fixed with 4% paraformaldehyde in prewarmed PBS and washed three times with PBS. The cells identified by confocal microscopy were relocated using the coordinate system position. Images were acquired using a Leica SP8 microscope with a ×63 (NA: 1.5) oil-immersion objective and Lightning setup (HyVolution super-resolution, Leica Microsystems) and processed by Fiji software. The percentage of cells with mtDNA release was assessed by visual inspection of noncolocalizing TFAM–RED and TOM20 signals. When one or more TFAM nucleoid signals did not overlap with TOM20 labeling, we counted this as released.

### mtDNA analysis by qPCR
For quantitative analysis of mtDNA release using qPCR, we stimulated apoptosis in MEFs and extracted the cytosolic fraction with digitonin-containing buffer as described above. Briefly, wt, BAX/BAK-DKO, MTCH2-KO MEF and MTCH2-KI MEFs were seeded into six-well plates and incubated for 24 h under cell culture conditions. Next, cells were treated with COMBO (10 µM ABT-737 and 10 µM S63845) and QVDA (20 µM) for up to 3 h. Cells were harvested and resuspended in permeabilization buffer (20 mM HEPES–KOH pH 7.5, 100 mM sucrose, 2.5 mM $MgCl_2$, 100 mM KCl, freshly added 0.025% (w/v) digitonin and protease inhibitor cocktail in PBS) for 10 min on ice. To remove the nuclear fraction, samples were centrifuged at 1,500$g$ for 5 min. Next, samples were centrifuged at 20,000$g$ for 15 min and supernatants were collected for DNA extraction. mtDNA was isolated from cultured cells using the DNeasy blood and tissue kit (Qiagen) according to the manufacturer's instructions and qPCR was performed using mtDNA primers (cytochrome b, Dloop, non-nuclear mitochondrial and 16S; sequences in Supplementary Table 1). For each independent sample, qPCR was performed in technical triplicates as described previously[72]. The average $C_t$ value of each individual primer at a concrete data point was obtained and the $\Delta C_t$ value was calculated with respect to time zero. In Fig. 6a, we plotted individual $\Delta C_t$ values and connected them with dashed lines to indicate the mtDNA release tendency over time. Next, we calculated the area under the curve of the individual curves of three independent experiments using Prism.

### Cell death assays
In situ cell death assays were performed using the IncuCyte bioimaging platform (Essen) at 37 °C 5% $CO_2$. A total of 2–4 images per well were captured, analyzed and averaged. Cell death was measured by the incorporation of DRAQ7 (599/644) (D15106, Thermo). In the experiments, the medium containing DRAQ7 with or without ABT-737 or COMBO was monitored for 24 h. Cell death was assessed using IncuCyte basic analysis software.

### Apoptosis-resistant cell survival and death experiments
Wt or MTCH2-KO HeLa cells were treated with COMBO (10 µM S63845 and 10 µM ABT-737 for 24 h. Next, cells were washed with fresh medium and maintained in culture with regular medium exchanges. For apoptosis-resistant cell survival, cells were cultured for 1 week and then imaged using the IncuCyte bioimaging platform (Essen) at 37 °C and 5% $CO_2$. Whole-well images were captured, exported and analyzed using in-house software for label-free live–dead cell detection (CellLocator version 1.0.1; https://github.com/MichaelVorndran/CellLocator/releases/tag/V1.0.1). The number of total cells alive was correlated with apoptosis-resistant cells. For apoptosis-resistant cell death, cells were seeded into a 96-well plate after 48 h and analyzed using the same imaging and analysis workflow described above.

### Mitochondrial crista analysis by electron microscopy
Mitochondria images were obtained as described previously[46] with some modifications. Briefly, U2OS, HeLa cells or MEFs were seeded on Aclar discs and cultivated for 24 h before apoptosis induction with 5 µM COMBO (5 µM ABT-737 and 5 µM S63845) with 20 µM QVDA for 3 h. Then, cells were fixed for 1 h at room temperature and 30 min at 4 °C in 4% formaldehyde with 2.5% sucrose and 100 mM $CaCl_2$ in HEPES buffer pH 7.4. Samples were washed three times with 0.1 M cacodylate buffer. Then, were incubated with 1% osmium tetroxide and 1% potassium ferricyanide in 0.1 M CaCo buffer with 1.25% sucrose for 30 min at 4 °C. After three 5-min washes with 0.1 M cacodylate buffer, samples were dehydrated using an ascending ethanol series (50%, 70%, 90% and 100%) for 7 min each at 4 °C. Next, samples were infiltrated with a mixture of 50% Epon/ethanol for 1 h, 66% Epon/ethanol for 2 h and pure Epon overnight at 4 °C. TAAB capsules were filled with Epon and cured for 48 h at 60 °C as described previously[73]. Ultrathin sections of 70 nm were obtained using an ultramicrotome (Leica Microsystems, UC6) and a 45° diamond knife (Diatome) and stained with 1.5% uranyl acetate for 15 min at 37 °C and 3% lead citrate solution for 4 min at room temperature. Images were acquired using a JEM2100 Plus transmission electron microscope (JEOL) operating at 80 kV equipped with a OneView 4K camera (Gatan). Image analysis for mitochondrial crista morphology was performed manually using Fiji. Crista organization in mitochondria were classified as no cristae when cristae were not present, abnormal when cristae were swollen or with abnormal distribution, normal when cristae showed the classical elongated shape with uniform distribution and mixed phenotype when cristae showed patterns of two or more of the first three categories.

### *Helicobacter pylori* infection

First, wt, BAX/BAK-DKO and MTCH2-KO AGS or HeLa cells were cultured in RPMI 1640 medium supplemented with 10% FCS and infected with *H. pylori* G27 strain at a multiplicity of infection of 100 for 18 h, as described previously[52]. For subcellular fractionation assays, cells were seeded onto 15-cm dishes ($10 \times 10^6$ cells on the day of infection). After infection, cells were washed once with cold PBS, scraped off and centrifuged at 4 °C and 500*g* for 5 min. The cells were then resuspended in extraction buffer (250 mM sucrose, 70 mM Tris and 100 µg ml$^{-1}$ digitonin), incubated for 1 min and centrifuged again at 4 °C, 10,000*g* for 3 min. The supernatant containing cytosolic proteins was analyzed by WB.

For the analysis of cellular extracts, $1.5 \times 10^5$ cells were seeded onto six-well plates 1 day before infection. After infection, cells were washed once with PBS, lysed in RIPA buffer and analyzed by WB.

For the caspase 3/7 activity assay, cells were seeded in duplicates ($1.5 \times 10^5$) on six-well plates 1 day before infection. After infection, cells were pooled and lysed with lysis buffer (9803, Cell Signaling); protein concentration was measured using the DC protein assay (5000116, Bio-Rad). From each sample, 10 µl of lysate was incubated with reaction buffer (90 µl of MDB buffer, 11 µM Ac-DEVD-AMC (4026262, Bachem), 100 µg ml$^{-1}$ BSA and 0.1% CHAPS) in triplicate according to the manufacturer's recommendations. Measurements were performed with a Spark 10M plate reader (Tecan) using an excitation wavelength of 380 nm and an emission wavelength of 460 nm.

### Lipidomics experiments

Extraction of lipids from isolated mitochondria and analysis of phosphatidylcholine (PC), phosphatidylethanolamine (PE), phosphatidylinositol (PI), phosphatidylserine (PS), phosphatidylglycerol (PG) and PA species by nano-electrospray ionization (ESI) MS/MS were performed as previously described[44]. Levels of CL species were determined by LC–ESI-HRMS. First, 10 µl of the lipid extract in methanol was loaded onto an Acquity Premier BEH Shield RP18 column (100 mm × 2.1 mm inner diameter, particle size: 1.7 µm; Waters) and CL species were detected using a Q Exactive HF quadrupole Orbitrap MS instrument (Thermo Scientific). The LC (Vanquish Horizon Binary UHPLC, Thermo Scientific) was operated at 50 °C and at a flow rate of 0.4 ml min$^{-1}$ with a mobile phase of 10 mM ammonium formate in acetonitrile and water 60:40 (v/v) (solvent A) and 10 mM ammonium formate in isopropanol and acetonitrile 90:10 (v/v) (solvent B). CL species were eluted with the following gradient: initial, 30% B; 0.5 min, 30% B; 4.5 min, 68% B; 20.5 min, 75% B; 21 min, 97% B; 24 min, 97% B; 24.5 min, 30% B; 28 min, 30% B[74].

The MS instrument was operated in negative ion mode. Full MS scans in the range of *m/z* 1,200–1,700 were acquired with a resolution of 120,000, an automatic gain control (AGC) target value of $1 \times 10^6$ and a maximum injection time (IT) of 200 ms. For structural confirmation, parallel reaction monitoring (PRM) was simultaneously applied to acquire MS/MS spectra of precursor ions related to common CL species on an inclusion list. PRM scans were performed with a resolution of 30,000, an AGC target value of $1 \times 10^5$, an IT of 60 ms, an isolation window of 1.5 *m/z*, an isolation offset of 0.0 *m/z* and stepped normalized collision energies of 40, 60 and 80. All spectrum data were collected in the profile mode. The ESI source was operated with flow rates for sheath gas, auxiliary gas and sweep gas of 55, 15 and 2, respectively. The spray voltage setting was 3.50 kV, the capillary temperature was 360 °C, the S-lens radiofrequency level was 50 and the auxiliary gas heater temperature was 380 °C.

The exact *m/z* traces of the internal standards and endogenous CL species were extracted and integrated using TraceFinder 5.1 (Thermo Scientific). The peak areas of the endogenous CL species were normalized to those of the internal standards. The normalized peak areas were then normalized to the protein content of the mitochondria suspension. One replica in the sample for LPA-treated (Fisher Scientific, BML-LP100) U2OS ΔBAK GFP–BAK cells lacking MTCH2 was removed

because the sample contained significantly reduced total lipid amount compared to the other samples.

### Quantification and statistical analysis

Statistical information of the experiments is provided in the respective figure legends. Graphs were plotted using Python, Prism and Origin. Statistical significance was tested as described in the figure legends. Unless indicated differently, data are representative of three independent experiments.

### Reporting summary

Further information on research design is available in the Nature Portfolio Reporting Summary linked to this article.

## Data availability

Proteomics data were deposited to the PRIDE repository (PXD055201) and lipidomics data are available from Zenodo (https://doi.org/10.5281/zenodo.19006330)[75]. Data and materials can be obtained from the corresponding author upon reasonable request. Source data are provided with this paper.

## Code availability

The code used for the stoichiometry analysis is available from GitHub (https://github.com/jaufdermauer/stoichiometry_analysis).

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

## Acknowledgements

We thank C. Jüngst, F. Babatz, K. Seidel and A. Schauss of the CECAD Imaging Facility, M. Franz-Wachtel and B. Macek from the Proteomics facility at the University of Tübingen, S. Brodesser of the CECAD Lipidomic facility and A. Sarembe, G. Zimmer and J. Benecke for technical support. We are thankful to A. Gross for providing the MTCH2-KO MEFs, to S. W. Tait for providing SMAC–GFP and TFAM–RED constructs and to W. Fischer and R. Haas for providing the *H. pylori* G27 strain. This project was funded by the European Research Council (grant agreements 817758 and 101195637) and partially by the Deutsche Forschungsgesellschaft (German Research foundation), through SFB1218 project no. 269925409, SFB1403 project no. 414786233, SFB1530 project no. 455784452 and TRR353 project no. 471011418. H.F.-R. acknowledges support by the Ikerbasque research

fellow grant; by the European Union's Horizon 2020 research and innovation programme, COFUND, H2020-MSCA-COFUND-2020-101034228-WOLFRAM2; by the Spanish research ministry, grant PID2022-141616OA-I00 funded by MICIU/AEI/10.13039/501100011033 and the European Regional Development Fund a way of making Europe (ERDF/EU); and the RyC grant (RYC2024-049204-I) funded by MICIU/AEI/10.13039/501100011033 and ESF+.

## Author contributions

H.F.-R., A.P.-B., J.A., S.D., L.H., M.N. and Y.C.-S. performed the research and analyzed the data. J.A. analyzed the stoichiometry data. A.P.-B., S.D. and C.Z. analyzed the proteomic data. P.N. and G.H. provided the *H. pylori* model and performed the related experiments. E.C. and T.L. contributed to the qPCR data generation and analysis. A.L.S. and J.Z. generated the materials. A.J.G.-S. conceptualized the project and supervised the research. H.F.-R. and A.J.G.-S. wrote the manuscript with the help of all other authors.

## Funding

## Competing interests

The authors declare no competing interests.

## Additional information

**Correspondence and requests for materials** should be addressed to Ana J. Garcia-Saez.

# Reporting Summary

## Statistics

For all statistical analyses, confirm that the following items are present in the figure legend, table legend, main text, or Methods section.

| n/a | Confirmed | |
|---|---|---|
| ☐ | ☒ | The exact sample size (*n*) for each experimental group/condition, given as a discrete number and unit of measurement |
| ☐ | ☒ | A statement on whether measurements were taken from distinct samples or whether the same sample was measured repeatedly |
| ☐ | ☒ | The statistical test(s) used AND whether they are one- or two-sided *Only common tests should be described solely by name; describe more complex techniques in the Methods section.* |
| ☒ | ☐ | A description of all covariates tested |
| ☐ | ☒ | A description of any assumptions or corrections, such as tests of normality and adjustment for multiple comparisons |
| ☐ | ☒ | A full description of the statistical parameters including central tendency (e.g. means) or other basic estimates (e.g. regression coefficient) AND variation (e.g. standard deviation) or associated estimates of uncertainty (e.g. confidence intervals) |
| ☐ | ☒ | For null hypothesis testing, the test statistic (e.g. *F*, *t*, *r*) with confidence intervals, effect sizes, degrees of freedom and *P* value noted *Give P values as exact values whenever suitable.* |
| ☒ | ☐ | For Bayesian analysis, information on the choice of priors and Markov chain Monte Carlo settings |
| ☒ | ☐ | For hierarchical and complex designs, identification of the appropriate level for tests and full reporting of outcomes |
| ☒ | ☐ | Estimates of effect sizes (e.g. Cohen's *d*, Pearson's *r*), indicating how they were calculated |

*Our web collection on statistics for biologists contains articles on many of the points above.*

## Software and code

Policy information about availability of computer code

| Data collection | For proteomic assays, peptide mixtures were analyzed on an Easy-nLC II coupled to an LTQ Orbitrap XL mass spectrometer. Lipidomic data were acquired using Analyst 1.7.3 software (SCIEX). For microscopy imaging we used Zeiss, Leica and JEOL software. Further details are within the manuscript. |
|---|---|
| Data analysis | Proteomic analysis: acquired MS spectra were processed with MaxQuant software package version 1.5.2.8 with integrated Andromeda search engine. Database search was performed against a target-decoy Homo sapiens database obtained from Uniprot.(more details in the manuscript). Proteomic data will be available in PRIDE upon publication (Project accession: PXD055201). Lipidomic data:Extraction of lipids from isolated mitochondria and analysis of PC, PE, PI, PS, PG, and PA species by Nano-Electrospray Ionization Tandem Mass Spectrometry were performed as in Dadsena et al 2024 Nature Communications. Lipidomic data are available in Zenodo (https://doi.org/10.5281/zenodo.19006330). Microscopy data was analysed using microscope-specific software and Fiji. Stoichiometry data: To analyze the stoichiometry data in a high throughput manner, a Python script was developed, that is available under https://github.com/jaufdermauer/stoichiometry_analysis (default main_dbd branch). The function of the code is explained in detail in the Materials & Methods section. The license and instructions on how to use together with a test dataset are in the repository. |

For manuscripts utilizing custom algorithms or software that are central to the research but not yet described in published literature, software must be made available to editors and reviewers. We strongly encourage code deposition in a community repository (e.g. GitHub). See the Nature Portfolio guidelines for submitting code & software for further information.

## Data

Policy information about availability of data

All manuscripts must include a data availability statement. This statement should provide the following information, where applicable:
- Accession codes, unique identifiers, or web links for publicly available datasets
- A description of any restrictions on data availability
- For clinical datasets or third party data, please ensure that the statement adheres to our policy

All data supporting the findings of this manuscript are available from the corresponding author upon reasonable request. Proteomic data will be available in PRIDE upon publication (Project accession: PXD055201). Lipidomic data is available in Zenodo under https://doi.org/10.5281/zenodo.19006330

## Research involving human participants, their data, or biological material

Policy information about studies with human participants or human data. See also policy information about sex, gender (identity/presentation), and sexual orientation and race, ethnicity and racism.

| | |
|---|---|
| Reporting on sex and gender | NA |
| Reporting on race, ethnicity, or other socially relevant groupings | NA |
| Population characteristics | NA |
| Recruitment | NA |
| Ethics oversight | NA |

Note that full information on the approval of the study protocol must also be provided in the manuscript.

# Field-specific reporting

Please select the one below that is the best fit for your research. If you are not sure, read the appropriate sections before making your selection.

☒ Life sciences  ☐ Behavioural & social sciences  ☐ Ecological, evolutionary & environmental sciences

For a reference copy of the document with all sections, see nature.com/documents/nr-reporting-summary-flat.pdf

# Life sciences study design

All studies must disclose on these points even when the disclosure is negative.

| | |
|---|---|
| Sample size | The design of the experimental conditions was based on previous knowledge from similar experiments were done in the field. Also, pilot experiments were carried out that indicated the sample sizes chosen were adequate for a robust phenotype. |
| Data exclusions | One replicate from the lipidomic assay was excluded due to insufficient total lipid content, as indicated in the manuscript. |
| Replication | All experiments were performed with at least two or three independent replicates. All figures present the means and standard deviations (SD). All experimental replicates successfully validated the experimental findings. |
| Randomization | The samples were randomized and allocated in a way that made sense for the flow of the study, considering factors such as time and logical progression, without any preferential treatment. |
| Blinding | The data analysis was performed without blinding |

# Reporting for specific materials, systems and methods

We require information from authors about some types of materials, experimental systems and methods used in many studies. Here, indicate whether each material, system or method listed is relevant to your study. If you are not sure if a list item applies to your research, read the appropriate section before selecting a response.

## Materials & experimental systems

| n/a | Involved in the study |
|---|---|
| ☐ | ☒ Antibodies |
| ☐ | ☒ Eukaryotic cell lines |
| ☒ | ☐ Palaeontology and archaeology |
| ☐ | ☒ Animals and other organisms |
| ☒ | ☐ Clinical data |
| ☒ | ☐ Dual use research of concern |
| ☒ | ☐ Plants |

## Methods

| n/a | Involved in the study |
|---|---|
| ☒ | ☐ ChIP-seq |
| ☒ | ☐ Flow cytometry |
| ☒ | ☐ MRI-based neuroimaging |

## Antibodies

| | |
|---|---|
| Antibodies used | The study utilized Rabbit polyclonal anti-BAX (Cat#2772; RRID: AB_10695870; Cell Signaling Technology), Rabbit monoclonal anti-BAX (D2E11) (Cat#5023; RRID: AB_10557411; Cell Signaling Technology), Rabbit polyclonal anti-GAPDH (Cat#ab9485; RRID: AB_307275; Abcam), Goat polyclonal anti-Rabbit IgG-HRP (Cat#111-035-003; RRID: AB_2313567; Jackson Immuno Research), Goat polyclonal anti-Mouse IgG-HRP (Cat#115-035-003; RRID: AB_10015289; Jackson Immuno Research), Goat polyclonal anti-Rabbit (AF633) (Cat#A-21070; RRID: AB_2535731; ThermoFisher), Rabbit monoclonal anti-Tom20 (D8T4N) (Cat#42406; RRID: AB_2687663; Cell Signaling Technology), Mouse anti-Cytc (Cat#556433; RRID: AB_396417; BD Biosciences), Mouse monoclonal anti-GAPDH (D4C6R) (Cat#97166; RRID: AB_2756824; Cell Signaling Technology), Rabbit monoclonal anti-pTBK1/NAK, (Ser172) (D52C2) (Cat#5483; RRID: AB_10693472; Cell Signaling Technology), Rabbit monoclonal anti-TBK1/NAK (D1B4) (Cat#3504; RRID: AB_2255663; Cell Signaling Technology), Rabbit monoclonal anti-STING (D2P2F) (Cat#13647; RRID: AB_2732796; Cell Signaling Technology), Mouse monoclonal anti-GFP (Cat#MA5-15256; RRID: AB_10979281; ThermoFisher), anti-mouse IG-HRP (Cat#115-035-166; RRID: AB_2338511; Dianova), anti-rabbit-HRP (Cat#A6667; RRID: AB_258307; Sigma), anti-GAPDH (Cat#MAB374; RRID: AB_2107445; Cell Signaling Technology), anti-MTCH2 (Cat#PA5-88873; RRID: AB_2805186; Thermo Fisher), anti-Phospho-Histone H2A.X (Cat#2577; RRID: AB_2118010; Cell Signaling Technology), anti-SMAC (Cat#15108; RRID: AB_721554; Cell Signaling Technology), anti-AF488-anti mouse (Cat#A-11008; RRID: AB_143165; ThermoFisher), anti-Bak (Cat#12105; RRID: AB_2716685; Cell Signaling Technology), anti-Dlp1 (Cat#611112; RRID: AB_398423; BD Biosciences), anti-Flag (Cat#F3165; RRID: AB_259529; Sigma), anti-Mtch2 (Cat#ab105527; RRID: AB_10866682; Abcam), anti-Streptavidin-AF555 (Cat#S32355; RRID: AB_2571525; ThermoFisher), anti-Streptavidin-HRP (Cat#21126; ThermoFisher), mouse monoclonal antibody against BAX 6A7 (Cat#MA5-14003; RRID: AB_10979735; ThermoFisher), anti-MTCH1 (Cat#PA5100201; RRID: AB_2815731; ThermoFisher), Rabbit polyclonal to mtTFA - Mitochondrial Marker (TFAM) (Cat#ab131607; RRID: AB_11154693; Abcam), β-Actin antibody (Cat#sc-47778; RRID: AB_626632; Santa Cruz), Anti-α-Tubulin antibody, Mouse monoclonal (Cat#T6199; RRID: AB_477583; Sigma), and BID Antibody (Mouse Specific) (Cat#2003S; RRID: AB_10694562; Cell Signaling Technology). |
| Validation | All antibodies were validated by the supplier. |

## Eukaryotic cell lines

Policy information about cell lines and Sex and Gender in Research

| | |
|---|---|
| Cell line source(s) | The experimental models used in the study included the following cell lines and strains:<br>U2OS wt (Gift from S. Tait, Glasgow; doi.org/10.15252/embj.201899238),<br>U2OS BAK -/- KO (U2OS BAK KO) (Gift from S. Tait, Glasgow; doi.org/10.15252/embj.201899238),<br>U2OS BAK -/- BAX -/- DKO (U2OS BAK DKO) (Gift from S. Tait, Glasgow; doi.org/10.15252/embj.201899238),<br>U2OS BAK -/- KO GFPBAK (U2OS BAK KO GFPBAK) (This manuscript; N/A),<br>U2OS BAK -/- KO GFPBAK (U2OS BAK KO GFPBAK) MTCH2 KO (This manuscript; N/A),<br>U2OS BAK -/- KO GFPBAK (U2OS BAK KO GFPBAK) MTCH1 KO (This manuscript; N/A),<br>U2OS BAK -/- BID -/- KO GFPBAK (U2OS BAK KO GFPBAK) BID KO (This manuscript; N/A),<br>U2OS BAK -/- BID -/- KO GFPBAK (U2OS BAK KO GFPBAK) BID KO MTCH2 KO (This manuscript; N/A),<br>U2OS NUP96-mEGFP (Available from CLS cell lines service GmbH; Cat#300174),<br>Hela wt (ATCC; Cat#CCL2),<br>Hela Bax-/- BAK -/- DKO (Georg Hacker lab; doi:10.15252/embj.2018100907),<br>Hela MTCH2 KO (This manuscript; N/A),<br>MEF wt (Gift from Atan Gross lab; doi:10.1038/s41467-018-07519-w),<br>MEF MTCH2 KO (Gift from Atan Gross lab; doi:10.1038/s41467-018-07519-w),<br>MEF MTCH2 KO MTCH2 KI (This manuscript; N/A),<br>AGS WT (Sigma Aldrich, 89090402-1VL),<br>AGS MTCH2 KO (This manuscript, N/A),<br>MEF BAX -/- BAK -/- KO (MEF BAX BAK DKO) (Gift from Villunger lab ;N/A),<br>AGS BAX -/- BAK -/- KO (AGS BAX/BAK DKO) (doi: 10.1038/s41418-022-01009-9; N/A). |
| Authentication | Cell lines generated in this study were validated by western blot. |
| Mycoplasma contamination | All cel lines used in this study were subjected to regular mycoplasma testing using a kit (MycoStrip,Invivogen). |
| Commonly misidentified lines (See ICLAC register) | none of the cell lines used are commonly misidentified lines. |

# Animals and other research organisms

Policy information about studies involving animals; ARRIVE guidelines recommended for reporting animal research, and Sex and Gender in Research

| | |
|---|---|
| Laboratory animals | NA |
| Wild animals | NA |
| Reporting on sex | NA |
| Field-collected samples | NA |
| Ethics oversight | NA |

Note that full information on the approval of the study protocol must also be provided in the manuscript.

# Plants

| | |
|---|---|
| Seed stocks | NA |
| Novel plant genotypes | NA |
| Authentication | NA |

