## [Peer Review File · Nature Structural & Molecular Biology]

MTCH2 promotes BAX and BAK self-assembly and apoptotic pore growth

Corresponding Author: Professor Ana García-Sáez

A version of this paper was originally rejected for publication by Nature Structural & Molecular Biology, however that decision was reconsidered after appeal by the authors.

Version 0:

Decision Letter:

14th Feb 2025

Dear Dr. García-Sáez,

Thank you for submitting your manuscript "MTCH2 promotes BAX and BAK self-assembly and apoptotic pore growth". I apologize for the delay in processing your manuscript, which resulted from difficulties in obtaining referees' reports. Nevertheless, the comments from the 3 reviewers who have evaluated your manuscript are below. Unfortunately, after carefully considering their comments, we cannot offer to publish your current manuscript in Nature Structural & Molecular Biology.

You will see that while the referees find the work potentially interesting, and Reviewer #3 in particular offers positive comments, the reviewers also raise concerns about the extent to which the data supports the conclusions about MTCH2 acting as a key regulator of the BAX/BAK pore in apoptosis, and its role being independent of tBID. As it stands, the strength of the conclusions which can be drawn is insufficient for us to consider the manuscript further.

However, as you know, we are interested in this area of research. If further experimentation, analysis, and revisions allow you to address the referees' concerns in full, we would be prepared to consider an appeal of our decision, on the condition that no related work is published in the interim or has been accepted in our journal. We are always open to scheduling a phone call if you have any questions about the appeal process or decision. Please note that, until we have the opportunity to read a revised manuscript in its entirety in an appeal, we cannot make a decision as to whether it will be sent back for peer review. We would not encourage an appeal without substantial new experimentation addressing all the reviewers' comments thoroughly.

I am sorry we could not be more positive on this occasion. I hope that you find the referees' comments useful in deciding how best to proceed.

Sincerely,
Kat

Katarzyna Ciazynska, PhD
(she/her)
Senior Editor
Nature Structural & Molecular Biology
<https://orcid.org/0000-0002-9899-2428>

Referee expertise:

Referee #1: mitochondrial biology

Referee #2: apoptosis

Referee #3: apoptosis, molecular, Bax/Bak

Reviewers' Comments:

Reviewer #1 (Remarks to the Author):

Activation of the TNF α /Fas/TRAIL-death receptor pathway leads to caspase-8 cleavage of pro-apoptotic BID to truncated tBID, which translocates to the mitochondria to activate BAX/BAK oligomerization, resulting in MOMP and apoptosis. It was previously reported that mitochondrial MTCH2 functions as a novel receptor-like protein for tBID, and that MTCH2 knockout, both in-vitro and in liver-conditional mice, hinders recruitment of tBID to mitochondria, the activation of BAX/BAK oligomerization, MOMP, and apoptosis (Zaltsman et al Nat Cell Biol 2010).

In this MS, Flores-Romero et al demonstrate that MTCH2 localizes nearby BAX/BAK assemblies in apoptotic cells, and that MTCH2 knockout cells exhibit delayed BAX/BAK oligomerization and MOMP, resulting in decreased mtDNA release. The authors also report that the addition of lysophosphatidic acid (LPA) partially rescues BAX/BAK assemblies in MTCH2 knockout cells. In addition, Flores-Romero et al demonstrate that MTCH2's insertase activity is not required for BAX/BAK assemblies, and that MTCH2 is required for sublethal apoptosis signaling.

Specific comments

1. The findings described in this MS regarding the role of MTCH2 in BAX/BAK oligomerization are not novel enough and do not contribute much beyond the findings reported in Zaltsman et al Nat Cell Biol 2010. Also, the possibility that MTCH2 directly activates BAX/BAK (not likely in my opinion; see below #2) is not novel enough, but I'm open to be proven otherwise.

2. Page 8, 2nd paragraph from top: Flores-Romero et al argue that their findings are novel since BAX/BAK oligomerization did not involve tBID in their system (BH3-mimetics that directly activate BAX/BAK by inhibiting the anti-apoptotic BCL-2 proteins, and BID cleavage was not detected). Thus, they suggest that BAX/BAK are activated directly by MTCH2 independent of tBID.

If this indeed the case then the authors should show:

- BID knockout has no effect on BAX/BAK oligomerization in their system
- MTCH2 directly interacts with BAX/BAK in intact membranes using protein crosslinkers (as reported for tBID and MTCH2)
- MTCH2 mutants that disrupt the putative MTCH2-BAX/BAK interaction fail to rescue the MTCH2 knockout phenotypes

3. Fig 5: Similar data to the MTCH2 knockout lipidomics data presented in this figure was previously reported (initially in bioRxiv 2023 doi: <https://doi.org/10.1101/2023.12.15.571941> and in its final format in Chourasia et al EMBO J PMID: 39753955). Also, the observed accumulation of lipid droplets in MTCH2 knockout cells (Fig S6), appeared in the above-mentioned paper. Thus, the duplicated data should be removed and the similar data should be referenced.

4. It was previously demonstrated that cardiolipin or MTCH2 can serve as tBID receptors during apoptosis (Raemy et al Cell Death Differ 2016). Since, the authors of the current MS show that the addition of LPA to cells increases the levels of cardiolipin in the MTCH2 knockout mitochondria (Fig 5B), it is tempting to speculate that the partial rescue is due to recruitment of tBID (even thus it is not detected by Western blot) or recruitment of a different BH3-only activator protein like BIM or PUMA. The authors should determine which of the options is the correct one.

Reviewer #2 (Remarks to the Author):

The work by Flores-Romero identified that MTCH2, a mitochondrial carrier homolog protein, is a key regulator of the apoptotic pore formation during programmed cell death. Using APEX2 proximity-dependent labeling coupled with mass spectrometry, they successfully discovered that MTCH2 promotes BAX and BAK supramolecular oligomerization at mitochondria during apoptosis. I want to highlight several novelties and innovations of this study, including (1) the discovery of MTCH2's specific role in regulating apoptotic pore growth, (2) linking MTCH2's function to mtDNA release and cGAS/STING pathway activation and h. pylori infection model. This work advances our understanding of apoptotic pore regulation. Most of the data presented in this manuscript is of high quality, well-controlled, carefully performed, and clearly presented so the audience can easily digest it. The scientific rigor is high.

Although this work is highly innovative, I have a significant concern that stops me from supporting an immediate publication where I question the significance of MTCH2 in regulating apoptosis and furthering its biological functions.

Based on Figure S2, MTCH2 KO cells have NO defects in apoptosis induced by COMBO BH3 mimetics. These data are very disappointing, suggesting that although MTCH2 may regulate the Bax/Bak pores, it is dispensable for apoptosis. I would

recommend that the authors test other apoptosis induction conditions, such as those from extrinsic pathways and death receptors. It would be hard to convince the cell death field that this work may have significant impacts unless MTCH2 significantly affects cell death dynamics (measured by Annexin-V % or DRAQ7 %).

Also, in Figure S2, although labeled as in MTCH2 KO, on WB, it is clearly a MTCH2 band. The authors will need to use the "clean" KO, not the "pooled KO" cells, to perform all these experiments.

I understand that the authors turned into using "sub-lethal" apoptosis to demonstrate the role of MTCH2, as no difference in "lethal" apoptosis has been observed. However, since MTCH2 regulates BAX/BAK, the most relevant model to study "sub-lethal" apoptosis would be the BH3 mimetics-induced "persistent cell model" (see PMID: 36055199). I strongly suggest that the authors try this model. This will significantly improve the potential impacts of this work and address my concerns.

In Fig. 7, the authors chose the H.p. infection model to test the "sub-lethal" apoptosis. The authors will need to provide the rationale for this choice; for now, this data seems peripheral to the main conclusion. Does MTCH2 really play a critical role in H.p. infection in vivo? This data is not solid enough as the current model the authors used here is highly artificial--HeLa cells are not the natural cell target that H.p. would infect. Why is DNA damage critical to H.p. infection? If the authors still decide to employ the H.p. infection model, a more pathological model will be needed.

Reviewer #3 (Remarks to the Author):

NSMB article 43803 by Flores-Romero and colleagues presents new data on the regulation of apoptotic pore dynamics by MTCH2, a mitochondrial protein the authors identified in an unbiased APEX2-MS proximity labeling proteomics experiment. The authors integrated complementary cell-based, biochemical, and microscopy approaches along with appropriate controls to carefully determine that MTCH2 regulates apoptotic pore assembly and growth most likely by modulating mitochondrial lipid composition, which is known to directly influences apoptotic poration (kinetics of assembly and pore size although these have not been investigated mechanistically in this manuscript being suggested as future direction). They show that MTCH2 regulates known pathological consequences of apoptotic pore engagement including released mtDNA-mediated STING/cGAS inflammatory signaling and minority MOMP DNA damage induced by H. pylori bacterial infection. The data are important in the field and broadly revealing how MTCH2 manipulation may modulate apoptotic poration mechanistically and pathologically.

I have a few minor comments for clarification and otherwise think this is a very nice study and well written paper:

1. I find it interesting that GFP-BAX- Δ C forms apoptotic foci in BAK and BAX expressing cells (Fig. 4A-C). Is this true also in BAK/BAK DKO cells? My question does not need to be answered experimentally if the authors have not done the experiment, but if they did it would be useful to include and explain if recruitment is dependent on endogenous BAK and BAX.

2. In the discussion the authors mention discovery of RNA metabolism proteins in proximity to apoptotic pores, but they do not speculate why this may be.

3. In the discussion the authors write:

"Our results uncover a direct role on the regulation of BAX and BAK assembly and thereby on the growth of the apoptotic pore, by promoting the incorporation of new monomers into apoptotic foci and by generating additional seeding points."

I would frame this less hypothetically as

"Our results uncover a tBID-independent role of MTCH2 in the regulation of BAX and BAK assembly into apoptotic pores."

4. In Figure 3 panels B-E the cell genotype should be labeled as in panel F if they are intended as such. Also correct, "C, D) Accumulation of BAK into discrete foci quantified as S.D. of the GFP signal (D, should be C) or GFP maximum signal (D) over time." And $p < 0.025$ and $p < 0.001$ vs $0,025$ and $0,001$.

** For Springer Nature Limited general information and news for authors, see <http://npg.nature.com/authors>.

Version 1:

Decision Letter:

18th Dec 2025

Dear Dr. García-Sáez,

Thank you for your letter concerning your manuscript "MTCH2 promotes BAX and BAK self-assembly and apoptotic pore growth". We have now had a chance to discuss the points you raised in detail, and we have decided to send your paper back to review.

Before we can proceed, **I would like to request some additional documents and data which we need to have on file for all revised manuscripts.** Please use the link below to provide the requested materials.

I provide details below - we ask that you upload an updated Reporting Summary, and provide relevant source data.

Please let me know if you have questions.

Once the manuscript is back with us, we will send it back to review as soon as possible. Please note, however, that due to upcoming winter holidays, this might take longer than usual. In this case, please feel free to take longer if you need in providing the materials, as we might not be able to move on with review before January.

REPORTING AND DATA AVAILABILITY: Please provide updated files as follows with your revision. As you already know, we put great emphasis on ensuring that the methods and statistics reported in our papers are correct and accurate.

1. **REPORTING SUMMARY:** Please provide an updated file:

This form is a dynamic 'smart pdf' and must be downloaded and completed in Adobe Reader.

2. **REPORTING OF STRUCTURAL DATA:** If there are additional or modified structures in the revision, please submit the corresponding PDB validation reports with your other files, and use the figshare integration system to provide access to maps, half-maps, and models.

Manuscripts reporting new structures should contain a table summarizing structural and refinement statistics. Please include these tables for new cryo-EM or -ET (<https://www.nature.com/documents/nr-tables-cryo-em.doc> and modifying this file for ET), NMR (<https://www.nature.com/documents/nr-tables-nmr.doc>) and X-ray crystallography (<https://www.nature.com/documents/nr-tables-xray.doc>). To facilitate assessment of the quality of the structural data, a stereo image of a portion of the electron density map (for crystallography papers) or of the superimposed lowest energy structures (>10; for NMR papers) should be provided with the submitted manuscript. If the reported structure represents a novel overall fold, a stereo image of the entire structure (as a backbone trace) should also be provided. For cryo-EM structures, a representative micrograph showing individual particles should be provided in the figures alongside a processing workflow (please see our Editorial for more information: <https://www.nature.com/articles/s41594-025-01567-9>).

DATA DEPOSITION: We require deposition of coordinates (and, in the case of crystal structures, structure factors) into the Protein Data Bank with the designation of immediate release upon publication (HPUB). Electron microscopy-derived density maps and coordinate data must be deposited in EMDb and released upon publication. Deposition and immediate release of NMR chemical shift assignments are requested. To avoid delays in publication, dataset accession numbers must be supplied with the revised manuscript and appropriate release dates must be indicated at the galley proof stage.

3. **REPORTING OF LIGHT MICROSCOPY DATA:** For any revision that includes light microscopy data, we ask our authors to please include a completed light microscopy reporting table

[https://www.nature.com/documents/Light_microscopy_reporting_table.xlsx] to ensure the methods are described thoroughly. The table will be available to reviewers and

ultimately published, should the manuscript be accepted at the journal. When submitting the revised version of your manuscript, please pay close attention to our href="https://www.nature.com/nature-portfolio/editorial-policies/image-integrity">Digital Image Integrity Guidelines.

4. **CODE AND COMPUTATIONAL WORK:** If newly developed, unpublished code was used in this work, it must be provided with the revision and access to it must be disclosed in the Code Availability Statement in the manuscript. Please also provide the completed software submission checklist. This form is a dynamic 'smart pdf' and must be downloaded and completed in Adobe Reader, instead of opening it in a web browser: <https://www.nature.com/documents/nr-software-policy.pdf>

If molecular dynamics (MD) simulations were performed, please refer to this Editorial and provide a completed MD simulations checklist with your revision: <https://www.nature.com/articles/s42003-023-04653-0>

Lastly, if it applies, please complete and upload the completed machine learning checklist with your submission for review: <https://www.nature.com/documents/machine-learning-checklist.pdf>

5. SOURCE GEL IMAGES: Unprocessed scans should be provided for all gels and western blots presented in figures and should be clearly labelled with the figure number in the file title. The source gel images should be uncropped, unmodified images, with molecular weight markers. Uncropped, unmodified gel images from existing experimental repeats, in addition to the data shown in the figures, can also be presented in source data. Please provide these files as PDF with your revision.

Please ensure that all control panels for gels and western blots are appropriately described as loading or IP controls. Lastly, please ensure that all images in the paper are checked for duplication of panels and for splicing of gel lanes.

6. SOURCE NUMERICAL DATA: we urge authors to provide, in tabular form in excel files, the data underlying the graphical representations used in figures. This is to further increase transparency in data reporting, as detailed in this editorial (<http://www.nature.com/nsmb/journal/v22/n10/full/nsmb.3110.html>). Please provide one excel file per figure max, with one panel per tab. When submitting files, the title field should indicate which figure the source data pertains to.

7. DATA AVAILABILITY: this journal strongly supports public availability of data. All data used in accepted papers should be available via a public data repository, or alternatively, as Supplementary Information. If data can only be shared on request, please explain why in your Data Availability Statement and in the correspondence with your editor.

For some data types, deposition in a public repository is mandatory as detailed below:

<https://www.nature.com/nature-research/editorial-policies/reporting-standards#availability-of-data>

EXTENDED DATA FIGURES

You can use the link below to be taken directly to the site and submit your manuscript:

Link Redacted

Sincerely,
Kat

Katarzyna Ciazynska, PhD
(she/her)
Senior Editor
Nature Structural & Molecular Biology
<https://orcid.org/0000-0002-9899-2428>

Version 2:

Decision Letter:

Our ref: NSMB-A50363B

27th Jan 2026

Dear Dr. García-Sáez,

Thank you for submitting your revised manuscript "MTCH2 promotes BAX and BAK self-assembly and apoptotic pore growth" (NSMB-A50363B). It has now been seen by the original referees and their comments are below. The reviewers find that the paper has improved in revision, and therefore we'll be happy in principle to publish it in Nature Structural & Molecular Biology, pending minor textual revisions to satisfy the referees' final requests and to comply with our editorial and formatting guidelines.

We are now performing detailed checks on your paper and will send you a checklist detailing our editorial and formatting requirements in about 2-3 weeks. Please do not upload the final materials and make any revisions until you receive this additional information from us.

Sincerely,

Katarzyna Ciazynska, PhD
(she/her)
Senior Editor
Nature Structural & Molecular Biology
<https://orcid.org/0000-0002-9899-2428>

Reviewer #1 (Remarks to the Author):

The authors have done a splendid job in addressing all my concerns.

The only comment I still have relates to their interpretation regarding the decrease in mitochondria lipids in the MTCH2-deleted cells. As demonstrated in the Choursia et al EMBO 2025 paper, MTCH2 KO results in a cellular starvation phenotype which leads to increased catabolism, including a strong decrease in membrane lipids. Thus, it is more likely that the decrease in mitochondria lipids in the MTCH2-deleted cells is due to the increase in membrane lipid catabolism than to the loose of MTCH2's lipid scramblase activity. In this respect, I think it is important that the authors discuss and cite the papers describing MTCH2's lipid scramblase activity.

Reviewer #2 (Remarks to the Author):

The MS has been significantly improved and is almost ready to be accepted. I only have one minor concern.

The authors mentioned the "persister cell phenotype". I would suggest removing those texts, as they (persister cell phenotype) are not really tested in this MS.

For example, in Figure 7C (new figure generated during revision), which basically suggests that the KO cells are more resistant to apoptosis, rather than suggesting they are persister cells. There is a serious experiment to be done if the authors really want to prove that those cells are "persisters": after 7 days, they should be sensitive to apoptosis, more sensitive to ferroptosis, and easier to metastasize. And the authors need to prove this process requires HRI and ATF4 etc. See PMID: 36055199.

If the authors decide to conduct more experiments to strengthen their conclusions, it should be encouraged.

Reviewer #3 (Remarks to the Author):

The revised manuscript further strengthens tBID-independent and integrase-independent role of MTHC2 in the regulation of MOMP kinetics. I have no further comments.

Version 3:

Decision Letter:

1st Apr 2026

Dear Dr. García-Sáez,

We are now happy to accept your revised paper "MTHC2 promotes BAX and BAK self-assembly and apoptotic pore growth" for publication as an Article in Nature Structural & Molecular Biology.

Your paper will be published online soon after we receive proof corrections and will appear in print in the next available issue. You can find out your date of online publication by contacting the production team shortly after sending your proof corrections.

If you have not already done so, we strongly recommend that you upload the step-by-step protocols used in this manuscript to the Protocol Exchange. Protocol Exchange is an open online resource that allows researchers to share their detailed experimental know-how. All uploaded protocols are made freely available, assigned DOIs for ease of citation and fully searchable through nature.com. Protocols can be linked to any publications in which they are used and will be linked to from your article. You can also establish a dedicated page to collect all your lab Protocols. By uploading your Protocols to

Protocol Exchange, you are enabling researchers to more readily reproduce or adapt the methodology you use, as well as increasing the visibility of your protocols and papers. Upload your Protocols at www.nature.com/protocolexchange/. Further information can be found at www.nature.com/protocolexchange/about.

Authors may need to take specific actions to achieve compliance with funder and institutional open access mandates. If your research is supported by a funder that requires immediate open access (e.g. according to [Plan S principles](https://www.springernature.com/gp/open-science/plan-s-compliance) or the [NIH public access policy](https://www.springernature.com/gp/open-science/us-federal-agency-compliance)) then you should select the gold OA route, and we will direct you to the compliant route where possible. Because authors warrant under our subscription licensing terms that they haven't committed to licensing any version of their article under a licence inconsistent with the terms of our agreement – including the applicable embargo period – publication under the subscription model isn't suitable for authors whose funders require no embargo.

Sincerely,

Katarzyna Ciazynska, PhD
(she/her)
Senior Editor
Nature Structural & Molecular Biology
<https://orcid.org/0000-0002-9899-2428>

Point-by-point answers to the reviewers' comments

Reviewers' Comments:

Reviewer #1 (Remarks to the Author):

Activation of the TNF α /Fas/TRAIL-death receptor pathway leads to caspase-8 cleavage of pro-apoptotic BID to truncated tBID, which translocates to the mitochondria to activate BAX/BAK oligomerization, resulting in MOMP and apoptosis. It was previously reported that mitochondrial MTCH2 functions as a novel receptor-like protein for tBID, and that MTCH2 knockout, both in-vitro and in liver-conditional mice, hinders recruitment of tBID to mitochondria, the activation of BAX/BAK oligomerization, MOMP, and apoptosis (Zaltsman et al Nat Cell Biol 2010).

In this MS, Flores-Romero et al demonstrate that MTCH2 localizes nearby BAX/BAK assemblies in apoptotic cells, and that MTCH2 knockout cells exhibit delayed BAX/BAK oligomerization and MOMP, resulting in decreased mtDNA release. The authors also report that the addition of lysophosphatidic acid (LPA) partially rescues BAX/BAK assemblies in MTCH2 knockout cells. In addition, Flores-Romero et al demonstrate that MTCH2's insertase activity is not required for BAX/BAK assemblies, and that MTCH2 is required for sublethal apoptosis signaling.

Specific comments

1. The findings described in this MS regarding the role of MTCH2 in BAX/BAK oligomerization are not novel enough and do not contribute much beyond the findings reported in Zaltsman et al Nat Cell Biol 2010. Also, the possibility that MTCH2 directly activates BAX/BAK (not likely in my opinion; see below #2) is not novel enough, but I'm open to be proven otherwise.

We thank the reviewer for raising this point and would like to bring up some arguments that we hope will convince them regarding the novelty of our study.

As the reviewer rightly mentions, Atan Gross and coworkers elegantly showed in their paper in Nat Cell Biol in 2010 that MTCH2 can function in recruitment of tBID to mitochondria, so that MTCH2 knock out hinders its binding to mitochondria during tBID-induced apoptosis, as well as cell death. In this study, the authors measured the levels of BAX and BAK activation during tBID-dependent apoptosis in wt and MTCH2 KO systems. Most importantly, the model proposed in the study is that MTCH2 regulates apoptosis by acting at the level of tBID recruitment to mitochondria (Figure R1). In this regard, and to put the role of tBID in perspective compared to BAX and BAK assembly, we would like to point out that tBID activates BAX and BAK to induce apoptosis under very specific conditions in type II cells upon death receptors activation. This is of physiological relevance in hepatocytes, as the BID KO mice indicate (DOI: 10.1038/23730). In contrast, under most settings BAX and BAK are activated by other BH3-only proteins, and they can even be activated in absence of BH3-only proteins, which underscores the importance of understanding both tBID-dependent and -independent MOMP.

While we do not contest these findings, our results uncover an additional function on MTCH2 on apoptosis regulation by acting at the level of BAX and BAK high order oligomerization (Figure R1), which is conceptually different of BAX and BAK activation dependent on tBID mitochondrial recruitment proposed in Zaltsman et al, and which is physiologically relevant under different conditions. Our results reveal that MTCH2 can also regulate MOMP at the level of BAX and BAK macromolecular assembly (which is a step in the process of apoptotic pore formation distinct from and downstream of the activation step) and thereby the growth of the apoptotic pore, independently of BID (see below). We also show that this has implications that go beyond the classical apoptosis signaling, because it hinders the kinetics of release of large mitochondrial content, such as mtDNA, and the downstream activation of the STING pathway under low caspase activity, and that it has also consequences for minority MOMP in a bacterial infection model. In the revised version, we also show now relevance in the context of persister cancer cells.

The novelty underlying the discovery of a regulatory mechanism of BAX/BAK supramolecular assembly by MTCH2 is following: So far we understand very little about the regulation of the apoptotic pore itself, once it is formed and beyond the initial step of BAX and BAK activation. Only recently it has been recognized that BAX and BAK oligomerization is a dynamic process leading to an apoptotic pore that continues to grow during apoptosis progression, determining the temporal release of mitochondrial content and the induction of downstream inflammatory signaling. Our findings that MTCH2 regulates BAX and BAK high-order assembly are novel because they reveal that the apoptotic pore growth can indeed be regulated by mitochondrial proteins and uncover MTCH2 as a key regulator. The model arising from our discoveries thus advances that proposed by Gross and colleagues and provides new insight into a so far poorly understood aspect of MOMP, which is the regulation of the apoptotic pore dynamics (Figure R1). Our results also disentangle BAX/BAK activation and subsequent mitochondrial potential loss from their high order assembly leading mtDNA release.

Figure R1. Levels of MTCH2 regulation of apoptosis. Zaltsman et al. showed that MTCH2 promotes tBID recruitment to mitochondria, which once there, induces BAX and BAK activation. In our study, we show that MTCH2 regulates high-order assembly of the apoptotic pore formed by BAX and BAK, with consequences for mtDNA release and STING activation, among other.

2. Page 8, 2nd paragraph from top: Flores-Romero et al argue that their findings are novel since BAX/BAK oligomerization did not involve tBID in their system (BH3-mimetics that directly activate BAX/BAK by inhibiting the anti-apoptotic BCL-2

proteins, and BID cleavage was not detected). Thus, they suggest that BAX/BAK are activated directly by MTCH2 independent of tBID.

We apologize for the confusion. We would like to clarify that we do not propose that BAX/BAK are activated directly by MTCH2, but that MTCH2 contributes to BAX/BAK high order oligomerization and apoptotic pore growth after activation (see figure R1). Our experimental conditions are independent of tBID, but BAX/BAK activation is still triggered by treatment with BH3-mimetics, which blocks the inhibitory function of pro-survival BCL-2 proteins, leading to BAX/BAK activation. We have explained this better in the revised manuscript (p.8).

If this indeed the case then the authors should show:

- BID knockout has no effect on BAX/BAK oligomerization in their system

We thank the reviewer for raising this point. We have now performed experiments of BAK oligomerization in BID KO cells lacking or not MTCH2. As shown in new Suppl. Figure 2D-G, our results demonstrate that BID deficiency does not alter the delay in BAK foci formation and oligomerization kinetics upon MTCH2 KO, neither the rescue by MTCH2 reconstitution. These new data strengthen our results and demonstrate that the effect of MTCH2 on BAX/BAK high order assembly is independent of its effect on tBID.

- MTCH2 directly interacts with BAX/BAK in intact membranes using protein crosslinkers (as reported for tBID and MTCH2)

In our manuscript we provide evidence regarding the close proximity between active BAX and MTCH2 by proximity ligation assay (PLA), in addition to the unbiased APEX2 experiments. We think that the PLA approach is cleaner and stronger than protein crosslinking in the sense that the experiment is performed in situ, providing spatial information of the proximity between BAX/BAK and MTCH2. While these data indicate close proximity between BAX/BAK and MTCH2 during apoptosis, our results suggest that the mechanism by which MTCH2 regulates BAX/BAK high order assembly is rather through its function in mitochondrial lipid fluxes. Hence, the resulting model that we are proposing does not require a direct interaction between MTCH2 and BAX/BAK. We have explained this more clearly in the revised manuscript (p.13).

- MTCH2 mutants that disrupt the putative MTCH2-BAX/BAK interaction fail to rescue the MTCH2 knockout phenotypes.

As mentioned above, the mechanistic model derived from our experimental evidence does not require direct interaction between MTCH2 and BAX/BAK to explain the regulatory role of MTCH2 on BAX/BAK high order assembly.

Yet, to explore this possibility following the reviewer's suggestion, and since the structure of active BAX in the membrane is not known, we built a model for a potential MTCH2:BID complex with AlphaFold3 based on the hypothesis that a potential complex with BAX would involve similar interaction surfaces (given their sequence and structural similarity). We identified residue Y159 in the RESKY domain, previously reported as important for interactions with BID (DOI: 10.1074/jbc.M111.328377), as

potentially important for complex formation. We then designed mutants of MTCH2 at position Y159 and performed reconstitution experiments in U2OS Δ BAK cells stably expressing GFP-BAK and MTCH2 KO. All of the tested mutants could rescue the MTCH2 KO phenotype (Figure R2), in line with a non-interaction-based model of function and with the mechanistic differences between the impact of MTCH2 on tBID recruitment versus BAX and BAK oligomerization. Since these results are negative and we consider that they do not add much to our experimental evidence already including the results in BID KO cells, we decided to keep these data for review only, but are open to include them in the manuscript if the reviewer or editor think otherwise.

Figure R2. MTCH2 mutants in the RESKY domain rescue the delay in kinetics of BAK foci formation in MTCH2 KO U2OS Δ BAK cells stably expressing GFP-BAK.

3. Fig 5: Similar data to the MTCH2 knockout lipidomics data presented in this figure was previously reported (initially in bioRxiv 2023 doi: <https://doi.org/10.1101/2023.12.15.571941> and in its final format in Chourasia et al EMBO J PMID: 39753955). Also, the observed accumulation of lipid droplets in MTCH2 knockout cells (Fig S6), appeared in the above- mentioned paper. Thus, the duplicated data should be removed and the similar data should be referenced.

We thank the reviewer for bringing up the study by Chourasia et al. at EMBO J. This article was published just a couple of days prior to our submission and we had missed citing it, which we now do in the revised version of the manuscript. However, we would like to clarify that the lipidomics data are not duplicated. In Chourasia et al., the authors performed lipidomics analysis of MTCH2 KO cells, while we performed lipidomics of mitochondria isolated from MTCH2 KO cells. We find that analyzing the lipid composition of mitochondria and not of entire cells was relevant for our study, since cellular changes in total lipid content would not be sufficient to support conclusions about events taking place specifically at mitochondria. As such, we believe our data

constitute a valuable addition to work published by Chourasia et al. and should not be removed. We also believe that reproducing results is an important aspect of science. By including our data on lipid droplets as a supplementary figure we do not take value from the data shown in Chourasia et al. For these reasons, we decided to also maintain these results in the revised version, but we have now highlighted that they reproduce the findings by the Gross group (p.9).

4. It was previously demonstrated that cardiolipin or MTCH2 can serve as tBID receptors during apoptosis (Raemy et al Cell Death Differ 2016). Since, the authors of the current MS show that the addition of LPA to cells increases the levels of cardiolipin in the MTCH2 knockout mitochondria (Fig 5B), it is tempting to speculate that the partial rescue is due to recruitment of tBID (even though it is not detected by Western blot) or recruitment of a different BH3-only activator protein like BIM or PUMA. The authors should determine which of the options is the correct one.

We thank the reviewer for this suggestion. We now provide new experimental evidence that addition of LPA continues to rescue the effect of MTCH2 KO of BAK foci formation in cells lacking BID, demonstrating that this effect is independent of tBID recruitment. These new data are included in Suppl. Figure 2H-K.

Unlike for tBID, the binding of BIM and PUMA (as well as other BH3-only proteins) to membranes has been reported to depend on their hydrophobic C-terminal anchor (DOI:10.1042/BCJ20210352; DOI: 10.1093/emboj/17.2.384; DOI: 10.1074/jbc.M708814200; DOI: 10.7554/eLife.37689; DOI: 10.7554/eLife.88329). While we are not aware of any publication reporting that cardiolipin plays a role in the recruitment of these proteins to membranes, we have discussed this possibility in the revised text (p.13).

Reviewer #2 (Remarks to the Author):

The work by Flores-Romero identified that MTCH2, a mitochondrial carrier homolog protein, is a key regulator of the apoptotic pore formation during programmed cell death. Using APEX2 proximity-dependent labeling coupled with mass spectrometry, they successfully discovered that MTCH2 promotes BAX and BAK supramolecular oligomerization at mitochondria during apoptosis. I want to highlight several novelties and innovations of this study, including (1) the discovery of MTCH2's specific role in regulating apoptotic pore growth, (2) linking MTCH2's function to mtDNA release and cGAS/STING pathway activation and h. pylori infection model. This work advances our understanding of apoptotic pore regulation. Most of the data presented in this manuscript is of high quality, well-controlled, carefully performed, and clearly presented so the audience can easily digest it. The scientific rigor is high.

We thank the reviewer for acknowledging the high quality of our study.

Although this work is highly innovative, I have a significant concern that stops me from supporting an immediate publication where I question the significance of MTCH2 in regulating apoptosis and furthering its biological functions.

We thank the reviewer for acknowledging the highly innovative nature of our work. We address the reviewer's concern regarding the significance of our findings in the specific comments below.

Based on Figure S2, MTCH2 KO cells have NO defects in apoptosis induced by COMBO BH3 mimetics. These data are very disappointing, suggesting that although MTCH2 may regulate the BaxBak pores, it is dispensable for apoptosis. I would recommend that the authors test other apoptosis induction conditions, such as those from extrinsic pathways and death receptors. It would be hard to convince the cell death field that this work may have significant impacts unless MTCH2 significantly affects cell death dynamics (measured by Annexin-V % or DRAQ7 %).

We thank the reviewer for this suggestion, which led to important new results. We specifically avoided to use treatments inducing extrinsic apoptosis because of the already reported role of MTCH2 in regulating the recruitment of tBID to mitochondria and subsequent BAX and BAK activation. But following the reviewer's suggestion, we now explored the effect of MTCH2 KO on apoptosis sensitivity at different concentrations of BH3 mimetics in U2OS and HeLa cells. Remarkably, we could detect a decrease in cell death sensitivity (measured by DRAQ7 %) in MTCH2 KO cells compared to WT cells as we titrated down the COMBO treatment. These new results show that MTCH2 regulation of BAX/BAK pores indeed significantly affects cell death dynamics and are now included in Figure 7A.

Also, in Figure S2, although labeled as in MTCH2 KO, on WB, it is clearly a MTCH2 band. The authors will need to use the "clean" KO, not the "pooled KO" cells, to perform all these experiments.

Following the reviewer's suggestion, we have now generated "clean" MTCH2 KO HeLa cells (Suppl. Figure 6B), which we used in the experiments of apoptosis sensitivity (Figure 7A) and for the persister cell model (see below, Figure 7B,C).

I understand that the authors turned into using "sub-lethal" apoptosis to demonstrate the role of MTCH2, as no difference in "lethal" apoptosis has been observed. However, since MTCH2 regulates BAX/BAK, the most relevant model to study "sub-lethal" apoptosis would be the BH3 mimetics-induced "persistent cell model" (see PMID: 36055199). I strongly suggest that the authors try this model. This will significantly improve the potential impacts of this work and address my concerns.

We thank the reviewer for this suggestion, which also led to new and interesting results. We now performed experiments of persister cells comparing HeLa WT and MTCH2 KO and found an increased number of persister cells in MTCH2 KO cells, which additionally became more resistant to BH3 mimetic treatment, supporting a role of MTCH2 on the generation of the persister cell phenotype. These new data are shown in Figure 7B,C.

In Fig. 7, the authors chose the H.p. infection model to test the "sub-lethal" apoptosis. The authors will need to provide the rationale for this choice; for now, this data seems peripheral to the main conclusion. Does MTCH2 really play a critical role in H.p. infection in vivo? This data is not solid enough as the current model the authors used here is highly artificial--HeLa cells are not the natural cell target that H.p. would infect.

Why is DNA damage critical to H.p.infection? If the authors still decide to employ the H.p. infection model, a more pathological model will be needed.

We agree with the reviewer that using HeLa cells for H.p. infection was very artificial. Following the reviewer's suggestion, we have now repeated these experiments in gastric AGS cells, a cell line commonly used in the H.p. field. In line with an effect of MTCH2 on the sublethal mitochondrial permeabilization by BAX/BAK during bacterial infection, we found that MTCH2 KO reduced the extent of DNA damage. As the reviewer mentions, this is not critical for infection, but is used here as a proxy of sublethal MOMP (which causes low levels of caspase 3 activation, leading to partial ICAD activity that causes DNA damage, as reported in doi: 10.1038/s41418-022-01009-9). These data are now shown in Figure 7D,E and Suppl. Figure 6C. The former data with H.p. infection in HeLa cells is now provided in Suppl. Figure 6D,E.

Reviewer #3 (Remarks to the Author):

NSMB article 43803 by Flores-Romero and colleagues presents new data on the regulation of apoptotic pore dynamics by MTCH2, a mitochondrial protein the authors identified in an unbiased APEX2-MS proximity labeling proteomics experiment. The authors integrated complementary cell-based, biochemical, and microscopy approaches along with appropriate controls to carefully determine that MTCH2 regulates apoptotic pore assembly and growth most likely by modulating mitochondrial lipid composition, which is known to directly influences apoptotic poration (kinetics of assembly and pore size although these have not been investigated mechanistically in this manuscript being suggested as future direction). They show that MTCH2 regulates known pathological consequences of apoptotic pore engagement including released mtDNA-mediated STING/cGAS inflammatory signaling and minority MOMP DNA damage induced by H. pylori bacterial infection. The data are important in the field and broadly revealing how MTCH2 manipulation may modulate apoptotic poration mechanistically and pathologically.

We thank the reviewer for acknowledging the importance and novelty of our findings.

I have a few minor comments for clarification and otherwise think this is a very nice study and well written paper:

1. I find it interesting that GFP-BAX- Δ C forms apoptotic foci in BAK and BAX expressing cells (Fig. 4A-C). Is this true also in BAK/BAX DKO cells? My question does not need to be answered experimentally if the authors have not done the experiment, but if they did it would be useful to include and explain if recruitment is dependent on endogenous BAK and BAX.

We thank the reviewer for this comment. While this experiment is very interesting and worth performing, we feel that it falls out of the scope of this study, and we focused on the experiments requested by the other reviewers. Based on our previous results reported in Cosentino et al. (Mol Cell, 2022), we think that GFP-BAX- Δ C is capable of targeting mitochondria via recruitment through endogenous BAK once this is activated. Alternatively, it could target mitochondria via membrane insertion through the pore forming helices 5 and 6, since BAX- Δ C was used in initial studies to permeabilize

membranes (DOI: 10.1126/science.277.5324.370; DOI: 10.1021/bi036044c). We discuss these options in the revised text.

2. In the discussion the authors mention discovery of RNA metabolism proteins in proximity to apoptotic pores, but they do not speculate why this may be.

While RNA metabolism proteins are among the top hits identified in our study, we did not perform the experiments to validate these data, and therefore avoided making conclusions out of them. But we still thought it could be worth mentioning, in case someone has made similar observations or wants to follow up on these observation. Following the reviewer's comment, we have now speculated that RNA binding proteins might be related to mtRNA release in the discussion (p.12).

3. In the discussion the authors write:

“Our results uncover a direct role on the regulation of BAX and BAK assembly and thereby on the growth of the apoptotic pore, by promoting the incorporation of new monomers into apoptotic foci and by generating additional seeding points.”

I would frame this less hypothetically as “Our results uncover a tBID-independent role of MTCH2 in the regulation of BAX and BAK assembly into apoptotic pores.”

We thank the reviewer for this suggestion. We have modified the text accordingly (p.13).

4. In Figure 3 panels B-E the cell genotype should be labeled as in panel F if they are intended as such. Also correct , “C, D) Accumulation of BAK into discrete foci quantified as S.D. of the GFP signal (D, should be C) or GFP maximum signal (D) over time.” And $p < 0.025$ and $p < 0.001$ vs 0,025 and 0,001.

We have now relabeled the genotypes in panels B-E of figure 3 and corrected the caption as suggested by the reviewer.